# Nano-Based Vaccine Delivery Systems: Innovative Therapeutics Against Cancer and Neurological Disorders

**DOI:** 10.3390/ijms262110316

**Published:** 2025-10-23

**Authors:** Sarfraz Ahmed, David Gozal, Abdelnaby Khalyfa

**Affiliations:** 1Department of Biomedical Sciences, Joan C. Edwards School of Medicine, Marshall University, Huntington, WV 25755, USA; ahmeds@marshall.edu; 2Department of Pediatrics, Joan C. Edwards School of Medicine, Marshall University, Huntington, WV 25755, USA; gozal@marshall.edu

**Keywords:** nanoparticles, nano-based vaccines, immunotherapeutics, vaccine delivery systems, liposomes, nanocarriers, cancer, neurological disorders

## Abstract

Vaccines have emerged as one of the most effective biomedical strategies for the eradication of diseases. However, a significant limitation remains in their ability to induce comprehensive humoral and cellular immune responses. Recently, nanoparticles (NPs) have been advanced as a novel vaccine delivery approach to address reduced immunogenicity. Several nanoparticle-based agents have now been approved for human use, and NP-based formulations have shown remarkable potential to enhance immunogenicity and stability, supporting targeted delivery and controlled release either through co-encapsulation of adjuvants such as Toll-like receptor (TLR) agonists or the inherent immune-stimulatory properties of NP materials in minimizing cytotoxicity. Despite these advances, there remains a pressing need for vaccines capable of addressing complex and multifactorial diseases such as neurological disorders and cancer. Nanotechnology could be a viable solution to this challenge. The use of lipid-based NPs, particularly those encapsulating mRNA, has garnered attention for its adaptability in vaccine delivery. Current studies indicate that NP composition, surface charge and size may play a crucial role in modulating biodistribution, delivering immune-stimulatory molecules, targeting antigens and trafficking antigen-presenting cells (APCs), which enhance immune responses across mucosal and systemic tissues. This review highlights recent advancements in NP-based vaccines and delivery systems, and adjuvants for cancer and neurological disorders. The review also covers an overview of NP-based and alternative delivery systems, focusing on the mechanisms and innovations related to NP-based systems for immunotherapeutic applications in cancer and neurological disorders.

## 1. Introduction

Cancer and neurological disorders affect millions of people globally [1,2]. Recent advancements in cancer vaccines, targeting early antigens processing, hold the potential to enhance therapeutic and prophylactic efficacy against tumors, metastases and relapses, especially in patients with lower pre-existing anti-tumor immunity. However, current cancer vaccine approaches often fail to elicit sufficiently strong and targeted immune responses [3,4], highlighting the critical need for strategies that may efficiently deliver adjuvants and tumor antigens to antigen-presenting cells. Similarly, the delivery of vaccines in neurological disorders requires suitable delivery systems to cross the blood–brain barrier (BBB), which is regulated by a tightened interaction between brain microvascular endothelial cells (BMVECs) and other components of the neurocentral vascular system. While the BBB safeguards neurons from harmful entities, it also hinders the passage of immunogens, complicating the CNS-targeted vaccination [5,6]. Furthermore, current vaccines using recombinant proteins, DNA and peptides can potentially address current challenges such as risks from defective inactivation, safety concerns and preceding anti-vector immunity [7]. The creation of a long-lasting and efficacious immunization against multiple antigens, especially for a contagious agent or a target cell through the production of antibodies and CD8^+^ T cells is known as the vaccination pattern [7]. Through the utilization of pattern recognition receptors (PRRs), innate immune system cells identify antigens, antigens-associated molecular patterns and repetitive motifs. Antigen-presenting cells (APCs) and other cells that recognize antigens first are frequently found to have PRRs. However, their notable shortcomings are short-term immunity and weak immunogenicity [7]. This also entails the generation of vaccines with immune-stimulating adjuvants, which may include delivery vehicles. For instance, emulsions and nano- or micro-particles upgrade antigen (Ag) delivery and the activation of APCs and immuno-stimulators such as TLR—ligands that excite potent adaptive and innate immunity. Nonetheless, over the last few decades, frequent experimental adjuvants have been probed, and their clinical translation has been exponentially slow. Mineral salts containing aluminum as a main constituent, have been used since the 1930s as a major adjuvant in the United States [8]. These adjuvants are sufficient for evoking humoral immunity with tolerable safety profiles, but they lack the stimulation of immuno-stimulators of CD8^+^ and CD4^+^ T cell-dependent immunity. Hence, this limits their potential use in designing vaccines for intracellular interactions with antigens [8,9]. Thus, there is a critical need for innovative formulations and delivery systems that function as adjuvants, capable of eliciting robust and long-term cellular and humoral immune responses, while minimizing the virulence and toxicity typically associated with current vaccines. Recent advances in comprehending antigen presentation by innate immunogenic cells and their relationships to adaptive immunity have facilitated rational progress. In this context, the major histocompatibility complex class I (MHC-I), a membrane-bound protein complex that presents intracellular antigens to cytotoxic CD8^+^ T lymphocytes, plays a critical role in bridging adaptive and innate immune responses. These insights have yielded the design of nanoparticle-based vaccine delivery systems with the advent of advanced nanotechnology. Nanotechnology refers to the science related to NPs. NPs are described as tiny particles with a size of 10–100 nm and in some cases up to 1000 nm [10]. Nanoparticles can be classified into different categories based on composition, dimensionality, phase and dispersion characteristics. In general, NPs are usually classified into a wide range of domains; however, inorganic and organic NPs are the major classes, with multiple subclasses [11,12,13]. Among the major classes, biologically important subclasses may include polymeric nanoparticles, liposomes, dendrimers, nanogels, nanocrystals, micelles, emulsions, quantum dots and several others, offering novel advantages and therapeutic applications [11,12,13]. The physicochemical properties of NPs, such as shape, size, surface and charge, substantially influence in vitro and in vivo correlation by impacting physiological interactions and drug release kinetics, ultimately determining the efficacy and safety of nanoformulations in drug and vaccine delivery [14]. NPs with optimal physicochemical properties may promote cellular internalization and efficient endocytosis in vaccine delivery [15], while positive surface charges promote electrostatic interactions with cell membranes, increasing cellular uptake and immune response activation [16]. Surface modifications such as PEGylation in nanoparticles decrease immunogenicity, prolonging circulation time, while cholesterol integration may enhance membrane fusion and stability, making these properties pivotal for overcoming barriers in vaccine delivery [16]. Vaccines which utilize nanoscale carriers or NPs are referred to as nanovaccines and enhance antigen stability, specific delivery, controlled release and immune activation compared to current vaccines using inactivated pathogens or soluble antigens, which rely on the body’s natural uptake with no specialized delivery systems.

The principal elements of an available upgraded vaccine are (1) the response of adaptive immunity against an antigen (Ag); (2) the stimulation of the innate immune system by an immuno-potentiator; and (3) ensuring it is Ag targeting and immuno-stimulatory to APCs [9]. In this regard, NP-dependent vaccine delivery systems have been engineered to fulfill these criteria, which may have multi-fold advantages over current vaccines. These advantages include the following: (1) the prevention of Ag degradation and its stability by encapsulation in NPs; (2) the augmentation of the potency and immunogenicity of vaccines by the co-encapsulation of immune-stimulatory agents and Ag in NPs; (3) APCs can readily process and phagocytose the nano-based materials; (4) the elevation of cytotoxic T cell-dependent responses by nanoparticle-based cytosolic delivery of Ag, with the presentation of major histocompatibility complex—I and cross-presentation of Ag; (5) the crosslinking of the B cell receptor for the enhancement of humoral immunity by NP-dependent multivalent presentation of the Ag; and (6) the surface modification of NPs with steering ligands and functional moieties that permit cells and organ-specific targeting to APCs and lymphoid organs [17,18]. For the utilization of NPs as adjuvants or carriers, scientists aim to enhance the safety, efficacy and targeted delivery of specific vaccines [18,19].

As inferred from the aforementioned comments, one of the major advantages of NP utilization in vaccination is their ability to enhance Ag presentation to the immune system. NPs can prevent the Ags from being destroyed, extend their circulation time in the body and improve their uptake by APCs such as dendritic cells (DCs), leading to an ameliorated immune response, and enhanced production of antibodies or T cell responses against the target agents [20]. Additionally, the NPs can be functionalized to target specific tissues or immune cells, thereby increasing the efficacy of vaccine delivery. Surface modifications with antibodies or ligands empower the NPs to bind to specific receptors or immune cells, promoting their activation and uptake. This targeted approach upgrades the immune response while minimizing the potential adverse effects and requirements for high doses of Ag [18,20]. Furthermore, NPs can act as effective adjuvants for triggering and enhancing the immune response to the vaccine. By reducing toxic features and incorporating immune-stimulatory molecules, NPs can elicit a more robust immune response, allowing for lower vaccine doses and possibly reducing the number of required immunizations by improving vaccine compliance, and lowering costs. Moreover, the NPs can ensure the co-delivery of multiple biological agents such as Ags, immuno-modulatory molecules and adjuvants in a single dose of vaccine. This strategy of co-delivery boosts the synergistic effects among these components, leading to a balanced and more potent immune response [18,20]. The creation and use of Ag/adjuvant-containing NPs, and subsequent modifications in cancer immunotherapy and neurological disorders have grown rapidly and become incorporated into clinical settings where NPs displaying various Ags are co-delivered with adjuvants [21]. This review aims to provide a comprehensive overview of nanoparticle-based vaccines and delivery systems, emphasizing their mechanisms and therapeutic potential for cancer and neurological disorders. The review is based on studies from Google Scholar, PubMed, Scopus, Embase and Web of Science using keywords such as “nanoparticles,” “nanocarriers,” “nano delivery systems,” “nano-based vaccine,” “vaccine,” “cancer immunotherapy” and “neurological disorders.” The included studies focused on recent advances and key research in nanoparticle-based vaccine delivery systems for cancer or neurological disorders, emphasizing efficacy, immunogenicity and delivery efficiency. Here, in Section 1.1, Section 1.2, Section 1.3, Section 1.4, Section 1.5, Section 1.6 and Section 1.7, we will describe various nano-based vaccine delivery systems or biologically inspired nanocarriers outlined below, while Table 1 and Table 2 will summarize pre-clinical and clinical stage nano-based delivery platforms for vaccines for these sections.

### 1.1. Lipid NPs and Liposomes

During the last few years, nanovaccines have gained a lot of attention for their efficacy, vaccination strategies and targeted transfer to achieve desired immune responses. In the rapidly developing field of nanotechnology, solid lipid nanoparticles (SLNs) are at the forefront of these efforts. SLNs have an extensive range of potent implementations in the delivery of small molecules, vaccine components, nanomedicine, proteins and genes via multiple modes of administration [22]. The application of SLNs for vaccine delivery has proven to be an excellent method for creating vaccines since they have the potential to improve site-specific transfer, antigen presentation, innate immune response initiation and potent T cell response, overcoming the major issue of safety when used against cancer, autoimmune disorders and neurodegenerative diseases [22,23]. SLNs can be engineered to carry a positive surface charge, which facilitates their interaction with negatively charged cell membranes, enhancing their cellular uptake. This property makes them particularly suitable for DNA-based and RNA-based vaccine delivery, where efficient cellular entry is critical for the expression of the encoded antigens. Moreover, SLNs can directly target specific tissues, improving the precision and effectiveness of the vaccine. Recent studies have shown that SLNs can induce strong cellular and humoral immune responses, making them a promising platform for vaccine development against various ailments [24].

A lipid nanoparticle-based delivery system (LNP) is appropriate for mRNA encapsulation. It uses cationic lipids, helper phospholipids, cholesterol and polyethylene glycated lipids. The delivery system incorporates perfect mRNA complexity, lipid nanoparticle stability, the intracellular discharge of mRNA and a deterrence of non-specific interactions (Figure 1A). These revolutionary LNPs emerged with the success of mRNA vaccines during the COVID-19 pandemic. Upon administration, LNPs facilitate the delivery of mRNA into host cells, where it is translated into antigenic proteins, thereby eliciting a robust immune response. The incorporation of ionizable lipids has been particularly important, as this approach enhances endosomal escape, allowing the mRNA to reach the cytoplasm effectively. This is crucial for initiating the desired immune response and ensuring the effectiveness of the vaccine [25]. LNPs exhibit many advantages, such as the efficient encapsulation and compression of mRNA, stimulation of endosomal escape, enhancement of cellular uptake, prevention of mRNA degradation in the extracellular environment, legal approval for human usage and large-scale production [26].

Liposomes are one of the best lipid-based formulations for intranasal delivery. Their major advantages relative to others may include their efficacy in encapsulating conjugate complexes and the viability of simple alteration. These characteristics of liposomes promote enriched cellular and mucosal uptake with overall biocompatibility. In this delivery system, cationic liposomes can enhance mRNA-based vaccine usage so as to achieve a greater impact. Compared to typical parenteral approaches, intranasal immunization offers several benefits including enhanced immune responses, the convenience of administration and a decreased risk of needle-related side effects. By overcoming the difficulties involved in administering vaccinations by intranasal route such as mucociliary clearance and restricted antigen uptake, liposomes have shown promise as a means of improving vaccine efficacy [27,28]. Jamie and colleagues showed that a liposomal-based vaccine, which spatially segregates target and helper peptides, effectively induced a rapid and high-titer anti-ErbB2 response [29]. Similarly, a Phase I trial of Lipovaxin-MM demonstrated that a dendritic cell targeted liposomal vaccine against melanoma is safe for further clinical studies [30]. The versatility of lipid nanoparticles extends beyond mRNA and DNA vaccines. They have also been explored as delivery vehicles for protein- and peptide-based vaccines. By incorporating adjuvants and other immunomodulatory agents into the LNP formulation, it is possible to enhance the immunogenicity of these vaccines. For example, the addition of TLR agonists to LNPs can potentiate the activation of innate immunity, leading to a more robust and durable adaptive immune response. This approach has been shown to improve vaccine efficacy, particularly in cases where traditional vaccines have been less effective, such as in elderly populations or among individuals with compromised immune systems [31,32]. Thus, the use of lipid nanoparticles in vaccine delivery represents a significant advancement in modern medicine. Their capability to protect and efficiently deliver a wide range of antigens, coupled with their versatility and safety profile, makes them an ideal platform for the development of next-generation vaccines. As research in this field continues to evolve, it is expected that LNP-based vaccines will play a central role in combating a wide array of diseases, from infectious pathogens to chronic non-contagious conditions such as cancer and neurological disorders [22,25,31]. To place the role of LNPs into perspective, it is essential to compare them with viral vector-based systems, which have long been used as powerful tools for vaccine delivery but present their own unique challenges. Various viral vectors, including adenovirus, adeno-associated virus, alphavirus, cytomegalovirus, HBV, herpesvirus, lentivirus, measles virus, poliovirus, poxvirus, Sendai virus and vesicular stomatitis virus, have all been employed in both clinical and pre-clinical trials for their capability to enhance cellular immunity and provoke potent immune responses when included in vaccines [33]. However, these vectors pose toxicity challenges, particularly linked to endonuclease degradation, and also present other potentially serious adverse effects. In contrast, non-viral vectors contain DNA (usually plasmid DNA produced in bacteria) or RNA as an antigen, which is delivered into the target cell to initiate the targeted immune response. Apart from nucleic acids, LNPs like non-viral vectors delivering proteins and peptide antigens also show more potential for vaccine advancement. Nonetheless, these delivery platforms mitigate the risks associated with endonuclease degradation, exhibit enhanced efficacy, and virtually eliminate the need for repeated doses compared to naked DNA or viral vaccines [34]. Henceforth, the quest for non-viral vectors like LNPs emerges by seeking those that are capable of encapsulating or absorbing antigens and merging with the cell membrane to discharge them into the cell cytoplasm. Compared with viral vector immunization, LNP-like non-viral vector immunization presents numerous benefits. These include safety and efficacy due to the absence of viral components, the ability to accommodate DNA of unlimited size, large-scale production, minimal or negligible host immunogenicity, antigen safeguarding, targeting capabilities, sustained gene expression and adjunctive effects. Several reports indicate that cationic-based liposomes, when administered intranasally, exhibit greater absorption and enhanced bio-accessibility compared to their negatively charged counterparts [35,36]; for instance, research has employed surface-modified cationic liposomes as a delivery system to develop a nanovaccine utilizing the recombinant spike S1 subunit from SARS-CoV-2 [26,37]. Similarly, other studies conducted in pre-clinical and clinical models reported liposome-based formulations as delivery vectors for enhancing the efficacy of vaccine alternatives to viral vectors or their combinations with advanced modifications [35,36,37] (Table 1 and Table 2). The initial findings from these studies are encouraging for the future use of LNPs against cancer and neurological disorders. Nevertheless, we should point out that the usefulness of LNPs similar to viral vectors is also constrained by several drawbacks such as poor transfection efficiency, episomal expression, cellular toxicity and inflammation induced by unmethylated CpG DNA regions [38].

### 1.2. Polymeric NPs

Polymer-based NPs are considered an attractive option for a delivery system owing to their exclusive properties. The application of synthetic polymeric biodegradable NPs to deliver biomolecules has been investigated over the past two decades [39]. They have been used in biomedical applications involving drug therapies, in vivo antibiotic therapy and vaccines [40,41]. Devising these types of NPs is carried out by the configuration of several monomers as branched, linear and 3D networks so that their size, shape and surface charge can be simply modified. The cross-linkage of the polymer matrix permits the biomolecules’ encapsulation and delivery upon the degradation of the matrix (Figure 1B) [42,43,44]. Biodegradable polymers offer flexibility in the size of NPs, safety and controlled delivery of encapsulated biomolecules in non-targeted and targeted forms [40,41]. These nanoparticles can also be engineered to possess self-adjuvating properties, which eliminate the need for separate adjuvants. For example, the shape of polymeric nanoparticles plays a significant role in the efficacy of peptide-based vaccines. Studies have shown that rod-shaped nanoparticles can improve cellular uptake as well as antigen presentation compared to spherical nanoparticles, leading to a more robust immune response. This is particularly relevant in cancer immunotherapy, where the design of polymeric nanoparticles can influence the strength and type of immune response generated [45,46]. Among polymer-based NPs, chitosan is one of the first-rate formulations due to its non-toxic, biodegradable and biocompatible nature [47]. It exhibits cationic polymers and offers a variety of biomedical uses and special qualities like non-toxicity, biodegradability, biocompatibility and environmental friendliness. Hydrophilic polymer biomaterials having a three-dimensional cross-linked network structure are hydrogels based on chitosan. The processes for creating chitosan hydrogels are divided into two groups: cross-linking polymer chains produced chemically and physically. Because of their biological and physicochemical characteristics, these hydrogels are used in clinical settings. Because of their network structure that can contain pharmaceuticals and their ability to control drug release through temperature- and pH-responsive release techniques, chitosan hydrogels are useful in drug delivery or vaccines. Hydrogels’ mechanical qualities, porosity and swelling capacity also make them strong contenders for use as scaffolds during the development of new tissue [48,49]. Recent studies have indicated that chitosan-based nanoparticles enhance mucosal immunity when administered via the intranasal route. These nanoparticles can protect antigens from degradation while facilitating their uptake by antigen-presenting cells (APCs). The non-toxic, biodegradable and mucoadhesive characteristics of chitosan make it an excellent carrier for vaccines [50,51]. The dawn of the era of polymeric biodegradable NPs and adjuvant derivatives may potentiate the realization of the concept of a human vaccine for cancer and neurological disorders in the near future.

Recent advancements in polymeric nanoparticle technology have also focused on improving DNA vaccine delivery. By optimizing the synthesis of polymeric nanoparticles, researchers have been able to enhance the delivery and expression of DNA vaccines in target cells. For example, the use of acid-degradable polymeric NPs allows the encapsulation of DNA vaccines, ensuring their release in the acidic environment of endosomes, which promotes efficient gene expression and immune activation. This strategy has shown promise in cancer immunotherapy, where DNA vaccines can induce strong and long-lasting immune responses against tumor antigens [52,53]. Moreover, polymeric NPs are being studied for the delivery of mRNA vaccines to mucosal tissues such as the lungs. Biodegradable poly(amine-co-ester) (PACE) NPs have been optimized for mRNA delivery to the lungs, demonstrating their potential as a platform for mucosal vaccination. This approach could be particularly beneficial for the respiratory tract. By promoting local immune responses, these nanoparticles may enhance the protection provided by vaccines against diseases [54]. The development of polymeric nanoparticles for oral vaccine delivery is another area of active research. One of the main challenges with oral vaccines is the harsh environment of the gastrointestinal (GI) tract, which can degrade vaccine components before they reach their targets. To address this issue, researchers have developed polymeric nanoparticles that can prevent mRNA vaccines from degradation while ensuring their release in the GI tract. This strategy has the potential to overcome the limitations of current oral vaccines and improve their efficacy [55]. A study investigated polymeric nanoparticles targeting the Sialyl-Tn antigen, which has been observed to be overexpressed in gastric cancer cells. The results suggest that polymeric nanoparticles can effectively deliver antigens to tumor cells, enhancing immune response [56]. This study holds promise for the possible exploitation of polymeric nanoparticle-based vaccines against cancer of the lung, other types of cancers and neurological disorders in future.

### 1.3. Dendrimer NPs

Dendrimers are new hyperbranched globular nano-polymeric structures in three dimensions. Their appealing characteristics set them apart from other polymers on the market including their nanoscopic size, narrow polydispersity index, superior control over molecular structure, availability of several functional groups at the periphery and internal cavities as part of modifications. These NPs are symmetrical molecules, and they possess homogenous structures contrived in 3D highly branched networks, and non-polar agents encapsulated in their core. Dendrimers have been used in a wide range of fields of study. For those involved in the study of medication delivery, the potential use of dendrimers in drug and vaccine delivery is particularly exciting since they can serve as a platform for the coupling of the medication and the targeted moieties provided by terminal functions [57,58]. Furthermore, dendrimers’ adaptability can be increased by customizing their properties through the use of these peripheral functional groups [59]. Recent studies have demonstrated the potential use of dendrimers in delivering mRNA vaccines (Table 1 and Table 2). These can encapsulate and protect nucleic acids, enabling their safe transport into cells and ensuring effective gene expression. By leveraging their ability to form stable complexes with nucleic acids, dendrimers facilitate efficient cellular uptake and enhance the immunogenicity of RNA vaccines. This makes them a valuable platform for the next generation of vaccines [60], aimed at a broad range of diseases including viral infections, cancer and possibly brain disorders.

Dendrimers have been utilized to create innovative cancer vaccines. A notable example is the coordinative dendrimer-based nanovaccine, where dendrimers were coordinated with manganese ions to self-assemble with peptide antigens. This nanovaccine formulation demonstrated significant potential in enhancing antigen presentation and stimulating robust immune responses. The dendrimer’s ability to improve the delivery and presentation of tumor antigen positions makes it a powerful tool in cancer immunotherapy [61]. Another cutting-edge application of dendrimers in vaccination involves the development of photothermal-triggered nanovaccines. For instance, G5 poly-amidoamine (PAMAM) dendrimers have been employed to design nanovaccines that can be triggered by photothermal effects. When exposed to light, these dendrimer-based nanovaccines release their antigens, which are then efficiently processed by the immune system leading to a strong and targeted anti-tumor response. This approach not only enhances the effectiveness of the vaccine but also provides a controlled release mechanism that can be finely tuned for specific therapeutic needs [62]. Specific research is warranted for the utilization of dendrimer NPs for antigen presentation or vaccine development as vehicles against cancer and neurological disorders.

### 1.4. Micelles and Emulsion

Biocompatible and biodegradable, without surfactants, pickering emulsions were studied as hydrophobic medication delivery systems for the skin. Using all-trans retinol as a model hydrophobic medication, these are formulated from block copolymer nanoparticle (either poly(lactide)-block-poly(ethylene glycol) (PLA-b-PEG) or poly(caprolactone)-block-poly(ethylene glycol) (PCL-b-PEG))-stabilized emulsions of medium-chain triglyceride (MCT) oil droplets. These novel emulsions enable drug loading within either oil droplets or non-adsorbed block copolymer nanoparticles and oil droplets [63]. Using Franz cell technique on pig skin biopsies, retinol skin absorption was studied in vitro [64]. Further studies using confocal fluorescence microscopy made it possible to see how the Nile Red dye was absorbed by skin on histological sections. In contrast to the surfactant-based emulsion and an oil solution, the Pickering emulsions demonstrated a significant build-up of hydrophobic medicines in the stratum corneum. Loading within oil droplets and block copolymer nanoparticles improved medication absorption through the skin once again. This was attributed to the additional contribution or modification of free drug-loaded block copolymer nanoparticles. This phenomenon made it possible to adjust the drug’s administration to the skin over a broad range by choosing the right formulation or drug-loading technique [63]. Micelles are nano structures formed by the assembly of amphiphilic molecules such as surfactants or polymers in aqueous solutions. They have a hydrophilic outer layer and hydrophobic core. These are primarily used to encapsulate antigens or hydrophobic drugs to improve their stability and solubility, facilitating controlled release and enhancing immune responses. These delivery vehicles are utilized for mRNA delivery applications. For example, a conjugate system with micelle and branched poly ethylenimine-stearic acid has been applied in the form of an RNA-based vaccine [59]. A study reported glyceryl monooleate (MO)-based reverse micellar carriers for the transcutaneous delivery of antigens. This led to a significant inhibition of tumor growth [65]. Another study was conducted on developing a therapeutic cancer vaccine using biodegradable and biocompatible micelles. The vaccine enhances antigen presentation via the major histocompatibility class I pathway, stimulating cytotoxic T lymphocyte immune response [66]. The researcher developed an LPN system capable of delivering an mRNA vaccine to the brain via I.V. injection. They showed that the vaccine effectively crosses the blood–brain barrier (BBB), presenting a new way of treating neurological diseases, which may pave the way for a wide range of conditions like Alzheimer’s disease, brain cancer, amyotrophic lateral sclerosis and others [67]. Similarly, another study used surface group-modified LNPs which not only cross the BBB but also target a special type of neuronal cells [68]. Hence, LNP vehicles used in several pre-clinical and clinical studies present futuristic applications for other diseases such as cancer and neurological disorders (Table 1 and Table 2). Recently, self-micro-emulsifying drug delivery systems (SMEDDSs) and protein-based Pickering emulsion systems have emerged as promising agents for enhancing drug and vaccine delivery. SMEDDSs are lipid-based formulations that spontaneously form microemulsions upon contact with aqueous media, improving the solubility and bioavailability of poorly water-soluble drugs. SMEDDS are lipids-based formulations and spontaneously make microemulsions through contact with aqueous media, while protein-based Pickering emulsions are generated using solid particles like proteins at the water–oil interface [69,70]. Although the literature lacks studies investigating these systems as carrier platforms for vaccines of cancer and neurological disorders, their properties suggest that these can be used as potential carrier systems in the future. For instance, Guo et al. (2025) introduced novel Pickering emulsions stabilized via double-metal ion adjuvants as vaccine delivery systems to increase humoral and cellular immunity in a COVID-19 vaccine [71], thus paving the way for their future use in vaccines for cancer and neurological disorders.

### 1.5. Inorganic NPs

Compared to organic materials, certain inorganic nanoparticles, for instance, gold, silica and iron oxide nanoparticles, exhibit hydrophilicity, stability and biocompatibility when properly surface-modified [72]. These particles also offer unique optical or magnetic properties, a high surface area per unit volume and the ability to be functionalized with specific ligands to enhance targeting to molecules or cells [72]. In addition to their capacity to regulate the release profile of pharmaceuticals, inorganic nanoparticles shield pharmaceuticals from deterioration and can lower dosages and the frequency of administration, which significantly lowers the toxicity of pharmaceuticals—especially those used to treat cancer. The development of modified innovative materials has led to improved drug delivery systems with fewer side effects [72]. Other than calcium phosphates, recent nanotechnology developments have led to the introduction of other inorganic nanoparticles as effective vaccine delivery matrices. Nowadays, nanoparticles possess extremely sophisticated chemical characteristics and numerous inorganic nanoparticles have found applications as drug or vaccine carriers [73]. Engineered inorganic nano-structures are valuable tools owing to their specific ability to function as old-fashioned delivery systems. They have been optimized for intranasal delivery whereby gold nanoparticles play a critical role in stimulating the immune system [59]. In addition to gold nanoparticles, other inorganic nanoparticles like silica nanoparticles and quantum dots have been explored for vaccine delivery. Silica nanoparticles offer tunable pore sizes and large surface areas, which allow for high antigen loading and controlled release. Quantum dots, with their unique optical properties, enable simultaneous imaging and delivery, providing a dual function in both therapeutic and diagnostic applications. The integration of these nanomaterials into vaccine platforms offers promising avenues for the development of next-generation vaccines with improved efficacy and safety profiles [74]. The use of inorganic nanoparticles for cancer treatment and detection has been the subject of several studies, and their field applications are constantly expanding.

Inorganic nanoparticles have gained considerable attention as platforms for vaccine delivery due to their unique physicochemical properties. Recent studies have shown that among these inorganic NPs, the ability to associate antigens through modifications like adsorption via polyelectrolyte multilayers (PEM) is particularly noteworthy [74,75]. This “layer-by-layer” strategy enables precise control over antigen presentation, enhancing immune recognition and response. The modular nature of PEM assembly allows for the inclusion of multiple antigens or adjuvants within a single nanoparticle, broadening the potential for creating multivalent vaccines. This approach has been shown to increase the immunogenicity of antigens while providing sustained release, which is critical for the development of effective vaccines [74]. A study has reported that AuNPs can deliver cancer antigens and effectively enhance the immune response [76]. AuNPs were conjugated with red fluorescent protein (RFP) and CpG oligodeoxynucleotides showing their potential as a vaccination strategy, and possibly against cancer. A similar approach of antigen presentation can be adopted for neurological disorders.

### 1.6. Immune-Stimulatory Complexes

Recent advancements have shown immune-stimulatory complexes (ISCOMs) to be another type of vaccine delivery vehicle. These have been shown to have strong adjuvant properties in various clinical trials. These micelles range 40 nm in size and contain colloidal saponin, phospholipids (phosphatidylcholine or phosphatidylethanolamine) and cholesterol [77]. A versatile use of ISCOMs is to entrap antigen envelope proteins. Similarly, another vaccine delivery vehicle akin to ISCOMs is also available, but without an antigen protein. However, when an antigen is loaded in the later stages, hydrophilic antigens can be quickly entrapped. This type of complex is known as an empty ISCOM or ISCOMATRIX [78]. ISCOMs have a unique structure for their strong binding interaction with saponins and cholesterol. Using the matrix of ISCOMs, cholesterol is considered to be a major component or modification as phospholipids and mainly phosphatidylcholine or phosphatidylethanolamine [79,80]. Some studies reported that phospholipids alone may provide a loose ISCOM matrix as compared to cholesterol, facilitating amphipathic molecules in obtaining proteins [79]. Vaccines of ISCOMs or ISCOMATRIX are produced using various types of antigens, including for pathogens [81]. The synthesis of ISCOMs involves detergent removal of mixed micelles that constitute modified Quil A (60–70%), lipids (10–15%), an appropriate detergent and the antigen (5–20%). The concentration of detergent along with the techniques of dialysis or ultracentrifugation can be applied for the removal of detergent [82]. ISCOMs have been used as vaccines carriers, particularly in infections; however, they may hold a futuristic use against cancer and neurological disorders.

### 1.7. Exosomes as Vaccine Carriers

Extracellular vesicles (EVs) are membrane-bounded vesicles with a diameter of 1 or >1 nm generated by all types of cells. They are categorized based on their size, chemical contents and biogenesis routes. Based on biosynthesis, exosomes (30–150 nm) and microvesicles (100–1000 nm) are the two major types of EVs [83]. Exosomes become fused with recipient cells’ plasma membranes and are capable of releasing their bundled substances into the cytosol [13,84]. Exosome-based vaccines hold promise for extensive application in therapies in the near future due to their role in disease progress and the prevention of illness, and their ability to stimulate immunological responses in the host. Exosomes are comparable to viruses in terms of size, delivery mechanism, chemical make-up, biogenesis and growth, along with their ability to enter host cells. Thus, EVs can be excellent vaccine agents as per their stability, biodistribution, solubility and permeability. A safe vaccine always warrants this strategy. EVs have been investigated for toxicity and immunogenicity [85]. BALB/c mice did not demonstrate inflammation or liver toxicity through induction by EVs extracted from human embryonic Expi293F nephrons. Similarly, EVs sourced from CD81^+^/CD9^+^/CD63^+^ cells did not show any effect on mRNA expression level in HepG2 cells [86]. An in vivo investigation demonstrated the robust safety of EVs utilizing CD63^+^/TSG101+ EVs generated from human 293T cells, exhibiting no toxicity or immunological response. The primary characteristic of EV-based vaccines such as their capacity to induce minimal immunogenicity, indicates that EVs can be employed in the secure manufacturing of vaccines. EVs, in comparison to their alternative delivery agents such as LNPs or viral vectors, have advantages like maintaining naive antigen structure, and ready access to all types of organs through physiological fluids. Hence, they exhibit a very effective antigen presentation system and supreme biosafety for meeting the requirements for effective vaccine development [87]. Extensive study is required to ascertain their specific toxicity. Notwithstanding their potential stability, they can facilitate large-scale production and the efficient loading of both hydrophilic and hydrophobic molecules. An intravenous injection of extracellular vehicles (EVs) is the prevalent method for delivering drug- or vaccine-loaded exosomes to target cells. This approach impedes the accumulation of the drug or vaccine in tumor tissues, as the exosomes are rapidly cleared from the bloodstream and sequestered in the liver or spleen [88]. Other methods of delivery may include oral, intraperitoneal, subcutaneous and intranasal delivery. Exosomes loaded with drugs and vaccines can be administered via an intranasal route to the central nervous system [89]. However, more research is warranted to explore the roles of exosomes in drug and vaccine delivery against cancer and neurological disorders.

## 2. Mechanisms of Nanomaterials for Enhancement of Vaccines’ Efficacy

### 2.1. NPs and Vaccine Permeability and Efficacy at Tissue Level

Three types of barriers such as the skin barrier, mucosal barrier and blood–brain barrier (BBB) are points to address in nanomaterials. For the skin barrier, the transdermal drug delivery system has contributed to treating skin-related diseases over the past two decades [90]. It should be noted that certain NPs can play a significant role in transdermal delivery systems due to their ability to assist drugs in penetrating the stratum corneum, which is widely recognized as a major physical barrier. As part of nano-drug formulations, nano-emulsions have been shown to enhance drug permeability to overcome skin barriers [91], and similar mechanisms may be applicable in specific vaccine delivery contexts. In addition, increased understanding of nanoparticle penetration at the molecular level was reported by Gupta and Riaz in 2017. The authors investigated the permeability of dodecane thiol-coated hydrophobic gold NPs of varying sizes and surface charges through lipid membranes using coarse molecular dynamics simulations. They claimed that the actual surface charge and particle size of gold NPs contribute to their high permeability, making them useful for transdermal delivery under certain conditions [91]. This principle could potentially be applied to certain types of NPs in future vaccine development. For the mucosal barrier, the nanomaterials can encompass the retention duration of the gastrointestinal tract, concurrently escape the drug being destroyed by enzymes and subsequently improve the absorption rate of the drug by hydrophobic, electrostatic and polymer chain interactions [92]. Ji and co-workers formulated complexes of NPs and recombinant adenovirus coated with (1) cell-pervasive peptide TAT to upgrade the transduction efficiency and (2) PEG to supply a hydrophilic core that would avert entrapment in hydrophobic mucus. The reformed nano-complexes could perforate the mucus membrane more than their preceding work without nanomaterials [93]. This approach makes drug permeation in the central nervous system possible, which is important in light of the fact that sufferers with diseases, brain metastases or brain tumors are not uncommon. There are three main possible mechanisms by which nanomaterials can pass through the BBB (blood–brain barrier): (1) nanomaterials are utilized as a carrier for drug absorption through the cerebral capillary wall, extending the retention time of drugs at absorption sites and upgrading the concentration gradient of drugs outside and inside the blood vessels, which is beneficial for the drug entering the brain; (2) nanomaterials enhance the lipids’ solubility in the vascular endothelial cell membrane, and generate surface activity and loading to enhance the permeability of drugs; and (3) a vaccine-like drug-loaded nanomaterial is engulfed by vascular cerebral endothelial cells, releasing the drugs into the brain [94,95,96]. The size of nanomedicines, operating within the bandwidth of 10–200 nm, achieves not only enhanced physiological properties of reactivity, surface area, strength, sensitivity and stability [94,96,97,98,99], but also improved penetration properties across the BBB and deep into brain tissues [94,96,98,99,100], creating an excellent delivery mechanism capable of transporting therapeutic medicines to the brain [95,96,98,99,101,102]. As an example, a published designed liposome with four combinations of pharmaceuticals (procarbazine, carmustine, doxorubicin and temozolomide) for the treatment of glioma boasts 10-fold effectiveness in its delivery compared to other techniques in cell models [103]. Two more reports showed that curcumin-loaded NPs can enter the blood–brain barrier (BBB), validating how secure the NPs can be before future vaccines [103,104].

### 2.2. NPs and Vaccine Permeability and Efficacy at Cellular Level

NPs can combine with immune cells such as macrophages and DCs to stimulate innate immunity, which plays a significant role in activating the adaptive immune system. This stimulation produces a long-lasting and robust immune response to Ags that are dissimilar to the target Ags. However, NPs can be produced to target specific tissues or cells by functionalizing their outer surface with ligands that can bind to specific receptors exposed to target cells such as cancer cells. This system of delivery allows for the vaccine to target the looked-for site of action directly, resulting in ameliorated efficiency and fewer off-target effects [105].

There are two principal mechanisms of nanoparticle-based vaccine delivery: a lipoplex-based delivery system and peptide-mediated nucleic acid transfection. In the lipoplex-based delivery system, (1) cationic-lipid-based micelles, also called liposomes, make complexes with DNA to form lipoplexes. (2) These complexes obtain access by endocytosis, resulting in the biosynthesis of inverted double-layer inverted micelles like vesicles. (3) The endosomal wall potentially fragments, liberating DNA in the cytoplasm and possibly towards the nucleus, all during the maturation of lysosomes from the endosome. (4) The acquired DNA might result in the expression of genes in the nucleus. Otherwise, DNA may be degraded in the lysosome [105]. In a peptide-mediated delivery system, both non-covalent and covalent peptide–DNA complexes act the same way as a lipid-based delivery system. The positively charged peptides must have the potential to (1) tightly pack DNA into compact and small particles, (2) target it to cell surface receptors, (3) stimulate endosome-mediated outflow and (4) target the DNA to the nucleus for the expression of reporter genes [105].

Vaccines are delivered using living or non-living vectors like NPs, as illustrated in Figure 2A. The vaccine loading on NPs is the primary step of a nanoparticle-based vaccine delivery system. The Ag can be loaded by several methods, such as adsorption, entrapment, conjugation and encapsulation. Adsorption means the physical attachment of the vaccine to the NP’s surface, while the capture of the vaccine within the matrix of nanomaterials is entrapment. Conjugation is the attachment of the vaccine on the nanoparticle surface covalently. Encapsulation includes the entrapment of vaccines within the interior of NPs [106]. The above-revealed methods support the integrity and stability of vaccines during transport and storage because NPs can preserve the vaccine from degradation and upgrade their bioavailability. The selection of a perfect loading method is based on the type of nanoparticle, vaccine and required properties of the ultimate vaccine delivery system. By employing these methods, the nanomaterials can deliver the vaccines to targeted cells or tissues, effectively leading to an improved immune response [106].

The peptide vaccine delivery mechanism should target APCs like DCs either passively or actively. If consumed, it will prevent peptides from degradation by increasing the maturation of APCs and interacting with essential components of the innate immune system such as TLR [106]. Nanotechnology and nanomaterials could meet the aforementioned needs due to their unique properties. Nanomaterials can also be perfectly useful as carriers or adjuvants for a therapeutic or preventative vaccine. To speed up the development of multivalent vaccinations, NPs can also carry different antigenic compounds simultaneously. Nanotechnology may increase the biological activity of vaccinations [107]. Figure 2B illustrates the typical mode of operation of mRNA-based vaccines.

### 2.3. Enhancement of Vaccine Targeting

Targeting, which has been extensively explored or shown, is the most valuable characteristic of nanomaterials. Personalized treatment in nanomedicine involves the usage of nanomaterials as the drug’s effective delivery system and reformulating the drug as a molecule that only affects the target cell or sick tissues, keeping the normal cells from being harmed [108]. According to this ideal concept, antibodies should be loaded on NP surfaces that can bind to the membrane-embedded proteins in specially adapted afflicted cells enabling them to function as “missiles” [109,110]. Lymphatic system tropism, active targeting (by adaptation), passive targeting (engulfment by macrophages as foreign bodies) and physical targeting (by the encapsulation of magnetic material) are some positive aspects of the nanomaterials [111]. The utilization of targeted delivery is explicitly crucial in cancer treatment as cancerous cells often demonstrate specific surface markers that can be targeted by nanomaterials. By delivering a specific vaccine to cancerous cells, nanomaterials can activate an immune response against the cancerous cells while reducing mutilation to healthy cells [111].

### 2.4. Stimulation of Immune Responses

NPs can stimulate immunity by activating the innate immune system, which plays a crucial role in initiating the adaptive system of immunity. When nanomaterials enter the body, they interact with immune cells such as macrophages and DCs, resulting in facilitated Ag processing, presentation and stimulation of T cells, which are required for maintaining and inducing adaptive immunity [112]. Furthermore, NPs can initiate the chemokines and cytokine secretion needed to orchestrate the immune system [113]. By adopting these mechanisms, nanomaterials can enhance the immune system response to vaccine Ags, causing a long-lasting and more pronounced immune response. The stimulation of the immune system by NPs is based on the shape, size, composition and surface chemistry of the nanomaterials. Researchers can control these variables to produce NPs as carriers that effectively deliver the vaccine to desired tissues or cells while stimulating the immune system [111].

Nanomaterials manifest several significant physical and chemical features owing to the quantum captivity effect and high surface area. NPs can directly stimulate immune responses independently of vaccine delivery [111]. Sun and Xia (2016) [113] documented several mechanisms of immune responses which energetically relate to the following: (1) the depot effect, (2) the activation of NLRP3 inflammation, (3) the perturbation of the DC membrane, (4) the regulation of autophagy, (5) the targeting of lymph nodes, (6) the signaling of TLR, (7) the instigation of B cells, (8) the diversity of T cells, (9) the presentation of Ags, (10) the release of host DNA and (11) soluble mediators. Thus, the perception of molecular mechanisms regarding immune activation is integral in the rational design of concocted NPs for optimal and long-established immune-potentiating effects. NPs play a crucial role in directing towards the defense pathway. Moreover, we are compelled to pay attention to the potential adverse effects of nano-drugs caused by the magnified immune response. Recently, mast cell degranulation, allergy and other adverse effects of some nanomaterials such as titanium dioxide nanofiber [114], silver NPs [115] and silica NPs have been extensively studied to improve the therapeutic profile [116].

### 2.5. Augmentation of Vaccine Cellular Utilization

Nanomaterials as a novel drug delivery system exhibit a high surface area and small size. These can quickly bind to biomolecules inside the cells and at the surface. Furthermore, after slight modifications, the drug’s structure can be changed, which is advantageous for the induction of immune responses [117]. NPs for DC-like cells have been utilized to deliver encapsulated viral Ags, which prompt a more systematic uptake of specific soluble Ags [118]. To establish this principle, Xu et al. (2017) utilized a pulmonary surfactant monolayer (PSM), which serves as a primary obstacle to describe the transport mechanism of nanomaterials [119]. The authors evaluated the mechanism of interaction of inhaled NPs with the PSM. They found that the binding strength that drives transport can be invigorated by enhancing lipid tail length and coating density. Thus, the binding power is the crucial factor that enhances cellular uptake [119]. This principle was also studied in animal models. For this purpose, to probe *Leishmania panamensis*, mice were treated with empty NPs, free cytosine-phosphate-guanine (CpG) and NP-CPG. The NP-CpG, surprisingly, led to preferential and unanticipated cellular uptake at the injection site. Thus, this manifests that combined with nanomaterials, the vaccines might be efficacious [120] when their biological interactions, including avidin–biotin linkage, are fused with nanomaterials for enhanced cellular uptake [121]. For example, a para-methoxyamphetamine hydrogel functionalized with biotin eventually transforms into a stable nano-complex with antibodies bound to avidin, revealing its improved cellular uptake in cancer cells [122]. Additionally, electrically stimulated plasmonic gold nano-particles have been described to exhibit unique properties. Their electrical stimulation can drive dipole-like and vibrational oscillations, demonstrating the ability to disrupt nearby cell membranes. Thus, mice immunized with this methodology demonstrated up to 100-fold greater uptake than the control group, which did not contain NPs or NPs as vaccine carriers [90].

## 3. Advancement in Cancer Vaccine Development

### 3.1. Cancer or Tumor Peptide Vaccines

Tumor peptide vaccines have garnered significant attention due to their ability to elicit specific immune responses against tumor antigens, thereby providing a targeted approach to cancer immunotherapy. These vaccines work by stimulating cytotoxic T lymphocytes (CTLs) that specifically target and kill cancer cells expressing the corresponding tumor antigens. Research has focused on enhancing the efficacy of peptide vaccines through various strategies such as the use of multivalent peptides, conjugated peptides, fusion proteins and self-assembled peptides, all designed to improve antigen presentation, and T cell activation [123]. For instance, multivalent peptides can target multiple epitopes, increasing the likelihood of an effective immune response by overcoming the heterogeneity of tumor cells [123].

In addition, extending the duration of antigen presentation is crucial for inducing a robust and sustained immune response. Studies have shown that extended peptide vaccines, which are designed for prolonged antigen presentation, result in superior CTL immunity [124,125,126]. This prolonged presentation allows dendritic cells (DCs) to maintain antigen availability, thereby enhancing the recruitment and activation of T cells. Moreover, the incorporation of adjuvants, such as cytokines or toll-like receptor (TLR) agonists, can further potentiate the immune response by modulating the tumor microenvironment and promoting the infiltration of immune cells into the tumor site [124,125,126].

Although traditional therapeutic methods such as surgery, chemotherapy and radiotherapy have improved, a strategic combination of conventional therapy and anti-tumor immunotherapy is emerging as the new and preferred course of treatment for tumors. However, it appears difficult to extend and improve the survival rate of cancer patients [127,128]. Because tumors contain specific tumor Ags, research has revealed that the immune system can distinguish between malignant and healthy cells. Tumor peptide vaccines directly enhance or activate anti-tumor immunity to destroy and remove tumor cells by expressing immunogenic tumor Ags with the help of chemotactic agents, cytokines and other adjuvants. The immune response can be triggered despite the absence of immune suppression or autoimmune disease [129], and it also has a minimal risk of cancer [130]. However, there are also downsides including a brief half-life and low immunogenicity [131]. This kind of tumor vaccination can readily overcome immune tolerance and the MHC restriction, while insufficient medication bioavailability can also be addressed [117,127]. Tumor peptide vaccines should be prioritized for improved outcomes when using a drug delivery system based on nanotechnology. A tumor peptide-based vaccine using a nano-drug technology was already debated thirty years ago and produced excellent results.

To minimize the side effects of the therapy, the NPs must first be targeted specifically [132]. Additionally, certain NPs are easily modifiable to display outstanding qualities like enhanced drug absorption and stability. Additionally, they keep Ags from degrading, extend their function time and gradually boost the effectiveness of Ag absorption and distribution [133]. They act as an adjuvant in many ways, improving the therapeutic consequences and immune effects of tumor peptide vaccines. Furthermore, NPs could act on the tumor’s co-transmitter and immune-potentiator simultaneously (APCs). Assembling NPs as tumor peptide vaccine carriers, which act on the immune system, currently provides better prospects for immunization [134]. Figure 3A depicts the mechanism of immunotherapy based on an NP-dependent tumor peptide vaccine. Recent evidence has delineated the effective therapeutic potential of releasing our immune response for cancer treatment [135,136]. However, the weak immunogenicity of cancer-reducing vaccines, suboptimal consequences of adoptive T cell transfer-based immunotherapies and off-target reactions of immuno-therapeutics are major challenges in cancer immunotherapy. Advanced physical-biochemical properties of NPs are convenient in drug delivery platforms and may address these practical challenges [135,136].

The underlying mechanism of NP-based vaccination as an anticancer agent involves several key steps: (a) NPs are engineered to carry tumor-associated Ags (TAAgs), peptides or proteins derived from cancerous cells that can provoke an immune response. The NPs protect the cancer Ags from degradation, ensuring their stability and presentation to immune cells. The NPs are taken up by APCs such as macrophages, DCs or B cells. Once within the APCs, (a) the NPs release the TAAgs. (b) The processing of TAAgs within the APCs and their presentation on the cell surface leads to their complexation with MHC molecules. This process enables the immune system to recognize cancer-specific Ags as foreign entities. (c) The TAAg presentation on APCs triggers the stimulation of T cells, which involves a specific interaction between the TAAgs-MHC complex and T cell receptors displayed on APCs. (d) The NPs can include immune-stimulating molecules such as cytokines and Toll-like receptors; these molecules also activate T cells, promoting a more effective and more robust immune response. (e) The activated T cells undergo clonal expansion, leading to the release of many tumor-specific T cells; this expansion helps to amplify the immune response in contrast to cancerous cells. (f) Once expanded and activated, the T cells approach the tumor site, guided by adhesion molecules and chemokines, and (g) these tumor-specific T cells identify cancerous cells presenting the TAAgs on their surface by T cell receptor–MHC interactions. This identification leads to the degradation of cancerous cells through several mechanisms including the secretion of cytotoxic molecules and the concurrent activity of recruited immune cells [135,136,137,138].

Overall, the utilization of NPs in cancer vaccination imparts a wide platform for efficiently delivering immuno-stimulatory molecules and TAAgs, upgrading the immune response in contrast to cancerous cells at the molecular level. By activating and targeting specific immune cells, these vaccines hold promise for personalized and effective cancer immunotherapy [139,140]. Nanomaterial systems are used to boost the targeted delivery of tumor therapeutics and Ags against immune checkpoint molecules by augmenting the effectiveness of adoptive cell transfer therapies and activating the immune system via new immune-stimulatory molecules, as shown in Figure 3B [135]. Adjuvants loaded into nanomaterials via electrostatic and hydrophobic interactions can boost cancer Ags’ immunogenicity. New prospects in cancer immunotherapy include neo-Ag identification via tumor exome sequencing and neo-Ag utilization for “precision medicine” by making new cancer vaccines [141].

The employment of targeted delivery is specifically important in cancer treatment, as cancerous cells often exhibit specific surface markers that can be targeted by nanomaterials. By delivering a specific vaccine to cancerous cells, nanomaterials can activate an immune retort against the cancerous cells while minimizing innocent bystander damage to the surrounding healthy cells (Figure 4A).

### 3.2. Exosome-Based Nanovaccines Against Cancer

Clinical studies are being conducted using exosomes as carriers, which can also be used as vaccines carrying specific antigens. Exosomes can transport drugs or formulations to targeted sites. They can be made from mesenchymal stem cells or extracted from patient fluids containing miRNAs or mRNAs [142]. Exosomes generated from DCs were demonstrated to cause more powerful T cell activation. The DEX vaccines produced by DCs pulsed with IFN-g were investigated in phase II clinical trials for NSCLC patients. Only one patient had grade III hepatotoxicity. The vaccinations largely promoted NK cell activity instead of T cell responses towards cancer [142]. A non-randomized phase I/II clinical trial explored exosomes produced by DCs burst with SART1, a biomarker for esophageal squamous cell cancer, as a vaccination. Various additional exosome-based immunizations have also been reported in clinical studies. Exosomes derived from granulocyte-macrophage colony-stimulating factor (GM-CSF) and ascites (AEXs) were investigated in a phase I study as a possible immunotherapy for colorectal cancer. Patients treated with AEXs with GM-CSF demonstrated robust anti-tumor cytotoxic T-lymphocyte responses against the colorectal cancer biomarker carcinoembryonic antigen [143]. Malignant ascites can develop when colorectal cancer cells enlarge and seed the peritoneal cavities. Patients with cancerous ascites typically have a poor prognosis. However, treatment for colorectal cancer patients with ascites may benefit from collecting exosomes present in the fluid and stimulating an immune response against the cancer via exosomes derived from the ascites [144]. AEX is considered a safe, acceptable and non-toxic vaccination against cancer. Shengming Dai and colleagues discovered that AEXs of colorectal cancer patients might induce anticancer immunity, while GM-CSF as an adjuvant may considerably boost AEXs’ efficiency. In vivo delayed type hypersensitivity (DTH) testing demonstrated that AEXs alone are sufficient to activate systemic anti-AEX immunity, suggesting that AEXs are immunogenic on their own and may stimulate CD8^+^ CTLs towards tumor antigens like carcinoembryonic antigen (CEA). In both in vivo and in vitro studies, exosomes formed by heat-stressed CEA-positive tumor cells can start and increase an HLA-A*0201-restricted and CEA-specific CTL response because they can accumulate heat shock proteins (HSPs) and MHC-I molecules [144]. To induce CEA-specific CTL responses with HLA-A*0201 restriction, the coadministration of GM-CSF and AEX was more effective than AEX alone, confirming the importance of antigen presentation and T cell activation. Combining AEX with GM-CSF may improve the effectiveness of AEX vaccination, offering an alternative for colorectal cancer immunotherapy [144]. An exosome-based cancer vaccine combined with CpG oligodeoxynucleotides or double-stranded RNA can enhance host immune responses, especially in viral infections. In clinical studies, GM-CSF has been extensively used as an adjuvant and shows potential as a vaccine component [83]. These strategies can be applied in cancer vaccine development. Although primarily aimed at cancer, exosome-based vaccines have also been developed to treat various chronic disorders. However, further studies involving advanced animal models and larger clinical trials are necessary to better evaluate and support the future use of exosomes as carriers for vaccines of cancer.

## 4. Nano-Based Carrier Systems for Neurological Disorders

Delivering vaccines to the central nervous system (CNS) to treat neurological disorders is fraught with several significant challenges, primarily due to the unique structural and immune features of the CNS. The BBB is a highly regulated barrier that does not allow entry of most molecules, such as vaccines, into the CNS. The BBB is made of endothelial cells, astrocytes and pericytes and forms a major barrier to the delivery of drugs to the CNS. Overcoming such a barrier requires specialized procedures in the form of nanoparticle-based systems capable of improving the permeability or temporally opening the BBB [145]. Also, the CNS has been categorized as an immune-privileged site, which implies that it has a reduced immune reaction as opposed to other body structures. This is due to such issues as the BBB and low lymphatic drainage, and immune-modulating agents, such as microglia and astrocytes. This means that vaccines which are effective through the stimulation of strong immune responses peripherally may not be as effective in the CNS [146]. There is also the blood–cerebrospinal fluid, which further blocks the transport of drugs or vaccines to the brain, serving as a barrier to separate the cerebrospinal fluid and blood. Efflux transporters may play a pivotal role in regulating the entry and exit of solutes and drugs or vaccine Ags’ remains from the CNS [147]. Efficient targeting is crucial for CNS vaccines, as non-specific delivery could result in neuroinflammation or unintended damage to healthy neurons. Nanoparticles such as lipid and polymeric particles are being designed to carry specific ligands that can target receptors unique to CNS cells, thereby improving the precision of vaccine delivery [148]. Examples of polymeric nanoparticles that can penetrate the BBB include the following: (A) nanocapsules, (B) tween 80-coated PBCA nanoparticles, (C) pegylated nanospheres, (D) nanospheres coated with ligands and/or antibodies, (E) nanospheres coated with ligands and/or antibodies and (F) pegylated nanospheres with extra ligands and/or antibodies (Figure 4B). Unlike peripheral tissues, the CNS lacks a well-defined lymphatic system for the transport of molecules. Of note, recent research points to the existence of the glymphatic system, which may serve as an alternative pathway for delivery. However, whether glymphatics can serve as a delivery conduit remains a topic of investigation [149]. Furthermore, while vaccines aim to elicit an immune response, excessive activation in the CNS can lead to harmful neuro-inflammation or even autoimmune reactions. Maintaining a balance between stimulating a protective immune response and avoiding overactivation is critical in CNS-targeted vaccine development [150]. These challenges emphasize the need for innovative delivery systems and a deeper understanding of CNS immunology to make CNS-targeted vaccines viable for treating neurological disorders. Alzheimer’s disease (AD), Parkinson’s disease (PD) and Huntington’s disease (HD) stand out as the three most prevalent neurodegenerative disorders (NDs) [151]. According to a recent report by the World Health Organization, more than 1.5 billion people worldwide currently contend with neurological disorders, encompassing AD, PD, stroke, headaches, brain injuries, epilepsy, neuroinfectious diseases and multiple sclerosis [152]. The literature lacks studies demonstrating nano-based carriers for neurological disorders; however, a brief overview of recent pre-clinical and clinical studies has been presented in Table 1 and Table 2.

### 4.1. Alzheimer’s Disease

Alzheimer’s disease (AD) comes in the form of a multifactorial cognitive disorder that develops from multifactorial progression complicated by beta-amyloid (Aβ) plaques and neuroinflammation. Previous studies have delved into the use of active immunotherapy using Aβ peptide as the antigen, which also proved the successful prevention of Aβ buildup by the production of anti-Aβ antibodies. Also, the transfer of regulatory T (Treg) cells by adoption showed a decrease in neuroinflammation in the brain of AD patients by the release of anti-inflammatory cytokines [153]. To enhance therapeutic effectiveness against AD, both approaches are integrated by developing a nanovaccine. This innovative vaccine presents Aβ peptides as antigens while simultaneously promoting the differentiation of naïve T cells into Aβ-specific Treg cells. To drive therapeutic superiority over AD, the nanovaccine will be built out of the two methods mentioned. A novel nano-based vaccine was developed that presented Aβ peptides as antigens and at the same time induced the differentiation of naive T cells to Aβ-specific T regulatory cells in a mouse model [154]. This nanovaccine was composed of lipid nanoparticles loaded with amyloid-β (Aβ) peptides and rapamycin, demonstrating promise for nanovaccine against AD. Mungyojung and others worked on creating a nanovaccine that can be used in the treatment of AD, which consists of triggering the development of antibodies against the amyloid and the creation of specific anti-amyloid Treg cells [153]. This nanovaccine (LNP-R/A/βs) is composed of lipid nanoparticles with amyloid-beta peptides with rapamycin. By targeting multiple pathological factors, the combination of anti-amyloid-β antibody therapy with amyloid-β-specific Treg cell transfers results in commendable outcomes. This approach presents a good solution to the limitations of AD immunotherapy. The nanovaccine was able to deliver amyloid-β peptides and rapamycin to the dendritic cells efficiently, and in turn, induce the production of anti-amyloid-β-specific antibodies and amyloid-β-specific Treg cells [153]. As a result, this LNP-R/Aβ, which has a size of 232.7 ± 84.0 nm and falls between 50–200 nm and >200 nm, successfully elicited the expected CD4^+^ T cell and antibody responses. When combined, this LNP-R/Aβ size would be ideal for both the intended immunological responses and efficient uptake by DCs [153]. A very recent pre-clinical study developed an mRNA vaccine (AV-1959LR) targeting the N-terminal region of amyloid-β for Alzheimer’s disease prevention. The vaccine was encapsulated in LNPs and administered intramuscularly using a dose of 5–50 µg in mice and 100 µg in cynomolgus monkeys. Antigen mapping showed that antibodies targeted the amyloid-β (Aβ_1–11_) peptide (EFRH) region, sera-bound human amyloid plaques and non-fibrillar halo regions. Results showed that the LNP-encapsulated mRNA vaccine (AV-1959LR) was immunogenic, generating long-lasting and plaque-binding antibodies, which can be a promising preventive candidate for Alzheimer’s disease immunotherapy [155].

The antigen carried by LNPs may be modified to produce Treg cells and antibodies that are specific to other therapeutic targets of AD, including hyperphosphorylated tau and ApoE4 [156]. This enables us to make changes to the LNP-R/Aβ vaccine platform. Therefore, LNP-R/Aβ may offer an alternative treatment platform (in the treatment of AD) using distinct pathways. LNP-R/Aβ are some of the beneficial attributes of LNP-R/Aβ [156]. LNPs are capable of delivering rapamycin and antigens, which the U.S. FDA has already approved as carriers of a variety of medications including a COVID-19 vaccination. In clinical settings, rapamycin has also been used to help patients receiving renal transplants to prevent organ rejection [157]. Furthermore, they employed human Aβ peptide as an antigen, meaning that AD patients can start receiving treatment with LNP-R/Aβs as a vaccine right away. When Aβ is present in the blood, it can be harmful because it can lead to the formation of Aβ plaques and neuroinflammation. But with their LNP-R/Aβ, Aβ was encased in LNPs, and after intradermal injection, the majority of LNP-R/Aβs were absorbed by DCs, thereby preventing Aβ from leaking into the bloodstream. After being absorbed by DCs, only the epitope portion of Aβ is visible on DC surfaces. As a result, Aβ in LNP-R/Aβ would not produce the toxicity that is frequently seen with conventional Aβ peptide vaccines [158]. Figure 5 demonstrates the mechanism of action of LNP-Aβ as well as suggested therapeutic strategies using LNP-R/Aβ. Recently, studies employing nanobiotechnology have been carried out to address the shortcomings of traditional treatments for neurodegenerative diseases. Because LNP-R/Aβ uses the benefits of nanotechnology (using the nanocarrier to codelivery rapamycin and the Aβ antigen) for the treatment of AD, it may help improve nanobiotechnology. When combined, the production and application of LNP-R/Aβ are therapeutically feasible. In the meantime, LNP-R/Aβ did not show any organ toxicity in vivo or cytotoxicity in vitro. By changing the shape of antigens to the appropriate one (such α-synuclein in Parkinson’s disease), it can also be used for many inflammatory disorders [158]. This approach effectively clears amyloid-β plaques in the brain, reduces neuro-inflammation, prevents excessive immune responses mediated by T helper 1 cells and mitigates cognitive recovery compared to traditional amyloid-β vaccines. This innovative nanovaccine holds significant promise as a novel treatment option for Alzheimer’s disease.

### 4.2. Parkinson’s Disease (PD)

Nanovaccination offers a novel therapeutic approach for managing Parkinson’s disease (PD) by utilizing nanoparticles to enhance the delivery and effectiveness of antigens, adjuvants or genetic material. Unlike conventional treatments which focus on alleviating symptoms, nanovaccination seeks to address the underlying mechanisms of neurodegeneration by targeting immune responses. In PD, characterized by the progressive loss of dopaminergic neurons, nanovaccination could potentially provide neuroprotection or modulate the neuroinflammation that accelerates disease progression [158]. One promising aspect of nanovaccination is the use of nanoparticles such as lipid-based, polymeric or inorganic nanocarriers to encapsulate therapeutic agents like antigens that can prevent or delay PD onset. A key focus is delivering alpha-synuclein (α-syn) peptides to stimulate an immune response that clears misfolded proteins, central to the pathogenesis of PD. By doing so, nanovaccines may help to remove α-syn aggregates, which contribute to neuronal death and disease progression [123]. Another strategy involves using nanoparticles to deliver antigens or biomolecules that modulate the brain’s inflammatory response. Neuroinflammation, driven by overactive microglia, is a major factor in PD-related neuronal damage. Nanovaccines that can reduce this immune-mediated inflammation may protect neurons from degeneration and slow disease progression [150]. Recent clinical trials showed peptide-based nanovaccines against PD using synthetic Aβ_1–14_ B cell epitope peptides [159,160,161].

### 4.3. Huntington’s Disease

Trophic factors have the potential to stop or reduce the degradation of neurons in Huntington’s disease (HD), as has been observed in other neurodegenerative disorders. One of the neurotrophic factors that has been shown to counteract the behavioral and neuropathological effects of intrastriatal injections of excitotoxins like quinolinic acid, is NGF [162]. The intrastriatal infusion of quinolinic acid, an NMDA receptor agonist, is a widely recognized animal model for the condition. Menei developed microparticles releasing NGF, with an average size of approximately 25 µm, drawing from the research group’s prior expertise in encapsulating NGF within PLGA. These microparticles facilitated the controlled release of bioactive NGF in vitro for a duration of five weeks [162]. When comparing the group treated with NGF-releasing microspheres to the untreated control group, the former showed a 40% reduction in lesion size following quinolinic acid infusion into the striatum. Additionally, this research study demonstrated that mice with NGF-releasing microparticles exhibited noticeable neuronal protection in the lesioned area [162]. Nanoparticles or vaccine-based studies for HD are scarce. However, Menei’s study on microparticles paves the way forward for the future delivery of nano-drugs and development of nanovaccines for HD.

One study has explored lipid nanoparticles (LNPs) for their role in delivering small interfering RNA (siRNA) or DNA vaccines to suppress the expression of *mHTT*. One study demonstrated that LNPs effectively transported siRNA across the BBB, lowering *mHTT* levels in the brain. This decrease led to reduced neuroinflammation and increased neuron survival in HD mouse models, showing promise for mitigating the progression of HD [112]. Research into dendrimer-based vaccines showed that they could co-deliver antigens and adjuvants, enhancing the immune response against toxic huntingtin protein fragments. This approach significantly reduced protein aggregation and neuronal damage in experimental HD models [163].

### 4.4. Multiple Sclerosis

The CNS is rendered inoperable by multiple sclerosis (MS), which also damages and deactivates the insulating layers covering the brain and spinal cord nerve cells. Therefore, a variety of debilitating physical, emotional and psychiatric issues arise from the brain’s failure to transmit signals [164]. To manage the severity of the MS conditions and symptoms, a number of FDA-approved medications are available for clinical use including cladribine (Mavenclad), dimethyl fumarate (Tecfidera), diroximel fumarate (Vumerity), fingolimod (Gilenya), monomethyl fumarate (Bafiertam), ozanimod (Zeposia), siponimod (Mayzent), teriflunomide (Aubagio), peginterferon beta-1a (Plegridy), glatiramer acetate (Copaxone and Glatopa), peginterferon beta-1a (Plegridy), alemtuzumab (Lemtrada), mitoxantrone hydrochloride, natalizumab (Tysabri) and ocrelizumab (Ocrevus). Drug concentrations in plasma and bioavailability are inadequate, and many medications have reduced pharmacological actions [165]. This necessitates the development of nano-based vaccines and drugs.

While much focus has been on drug delivery, nanoparticles have also demonstrated significant potential in developing vaccines for MS. Nanovaccines offer the advantage of targeted delivery of autoantigens or immunomodulatory molecules to induce immune tolerance, which is critical in treating MS as it is an autoimmune disorder [166]. For instance, polymeric nanoparticles have been explored for delivering myelin peptides or antigens to restore immune tolerance and suppress the autoreactive immune response against myelin. These are the vaccines that are created using nanoparticles and they can curb the onset of MS by re-programming immune cells to lower inflammation and conserve myelin [166]. In addition, lipid nanoparticles (LNPs) are more likely to be used as vehicles of vaccine delivery in MS, as they can penetrate through the blood–brain barrier (BBB). Nanoparticles have the ability to target immune cells such as dendritic cells to alter the immune response in a more controlled, sustained way through functionalization. As an example, dendrimer-based nanoparticles filled with immunomodulatory drugs or myelin peptide have demonstrated some pre-clinical promise and stimulated immune tolerance, and decreased the extent of demyelination in animal models of MS [166].

The existing methodology towards drug delivery with the solid lipid nanoparticle (SLN) has proven to have the potential for increasing the efficacies of some drugs with regard to their application in treating MS. Greater drug movement across the biological BBB was established through the use of riluzole-loaded SLNs, which were prepared through micro-emulsion in vivo. It is noted that when a rat model of MS was used, the concentration of the medication in the brain using a nano approach was increased, which boosted its neuroprotective effects compared to the other organs [167]. A nanovaccine against MS could potentially be developed in the future using a similar method to the nanovaccines in LNP-R/A/beta aimed at AD; these demonstrated efficacy in alleviating neuroinflammation and inducing immune regulation. Nanovaccines constitute a promising, novel and futuristic immunotherapeutic solution to the control and eventual regression of MS, since the delivery of MS-relevant antigens or immunoregulators can partly be accomplished through the use of nanoparticles. The idea with these novel vaccine systems based on nanoparticles is to re-educate the immune system and yield long-term benefits through the minimization of autoreactive immune responses and myelin repair [168]. Intriguingly, a recent animal study has shown promising results for a PS-LNP (phosphatidylserine lipid nanoparticle)-based vaccine for multiple sclerosis with an improved immune response [169].

### 4.5. Sleep Apnea

One of the most prevalent sleeping disorders is obstructive sleep apnea (OSA), which involves episodes of recurrent upper airway obstruction during sleep. OSA is associated with a greater cerebrovascular/cardiovascular disease and mortality risk [170,171]. It can also result in cerebral hypoperfusion which is caused by repeated hypoxic events and blood pressure spikes, and thus provides alterations in cerebral vascular autoregulation [172,173,174,175]. Several transcranial Doppler studies found cerebral blood flow (CBF) reductions during sleep in OSA [172,173,174,175]. Studies have revealed some of the neurocognitive deficits that patients with OSA experience and these include deficits in theory of mind skills, non-verbal reasoning, attention, episodic memory, executive functioning and mental shifting [170,171,176,177]. Also, the performances turn out to be worse in short-term verbal memory, constructional ability, phonological fluency and ability to inhibit dominant, automatic reactions as well [170,171]. Treatment for OSA includes pressure airway therapy (PAP) [178]. Additional therapies to treat OSA are weight loss, an oral device, craniofacial surgery, electrical stimulation implants to stretch the upper airway and tracheostomy [179,180,181,182]. In recent developments, the first practical potential pharmacological treatment of OSA was identified as tirzepatide, a pharmacological treatment for obesity and type 2 diabetes and a drug that is purely used for its the weight reduction properties [183].

New studies indicate that there is a close association between OSA and the risk of acquiring Alzheimer’s disease (AD). The investigations indicate that people afflicted by Alzheimer’s disease have an increased possibility of OSA compared with age controls of the same age group [184]. In animal experiments, intermittent hypoxia (IH), which is found to be one of the important hallmark characteristics of OSA, has been shown to increase the amyloid-beta and tau in brain areas indicative of Alzheimer’s pathology [184]. IH in apnea spells during sleep is a possible cause of nerve cell damage and encourages clumping of amyloid-beta plaques. Disrupted sleep patterns impair the ability of the brain to clear amyloid-beta-like toxic proteins, potentially enhancing the neurodegenerative processes [185]. The LNP-R/Aβ vaccine, which targets amyloid-beta (Aβ) proteins in AD, may thus represent a novel approach to reduce toxic protein aggregation in neurodegenerative conditions, including the subset of OSA patients at risk for AD. This vaccine employs LNPs to deliver RNA, which encodes multiple antigens that stimulate the immune system to clear amyloid-beta deposits. Thus, a similar immunotherapeutic strategy could potentially be adapted for treating OSA, even using other specific pathological and or inflammatory markers of OSA progression.

As indicated, OSA is linked to oxidative stress, systemic inflammation and neurocognitive dysfunction. Targeting molecular markers of pro-inflammatory cytokines such as TNF-α, IL-6 or hypoxia-inducible factors (HIFs) through an RNA-based vaccine could ameliorate the inflammatory pathways of OSA. Precise delivery using LNPs could potentially mitigate the chronic inflammation associated with OSA, ultimately reducing the risk of the complications associated with OSA. The concept aligns with advancements in immunotherapy for metabolic and neurodegenerative disorders, as demonstrated by research on RNA vaccines such as the LNP-R/Aβ [157,186]. While immunotherapy has shown promise in treating various diseases, its applications in OSA are not currently being explored. Some researchers have investigated the impact of immunotherapy on circadian rhythms and sleep, noting that sleep and circadian rhythms may influence outcomes of immunotherapy; however, these findings do not directly translate into a vaccine or immunotherapy for OSA [187]. Notwithstanding, LNP-R/Aβ-like vaccines may hold strong promise for their future application in specific subsets of patients with OSA.

### 4.6. Brain Tumors

One innovative method of delivering a targeted anticancer medication without harming healthy and normal cells consists of using sophisticated nano-drug carrier technology. Numerous medications and their derivatives including paclitaxel and etoposide solid lipid nanoparticles (SLNs) have been studied for their potential to treat glioblastoma, a highly aggressive brain tumor [188]. Although there has been a lot of interest in the delivery of drugs and nanoparticles have also become efficient in the delivery of brain tumor drugs, the literature lacks reports for NP-based vaccines. Tumor antigens or peptides encapsulated with nanoparticles can activate the immune system and destroy the cancer cells, leaving normal ones intact [189]. Specifically, tumor-associated antigens can be loaded into dendrimers and inorganic nanoparticles so as to activate dendritic cells and result in a strong immune response [98]. Lee et al., 2015, developed an RNA nanoparticle-based system for the targeted delivery of siRNA (small interfering RNA) to glioblastomas in mice models, but it was not a vaccine-based approach. The system utilized a self-assembled RNA nanoparticle platform, which demonstrated efficient delivery and the effective silencing of the target gene. This has demonstrated potential in pre-clinical data for treating glioblastoma [190], which paved the way for future vaccine delivery. Recently, Mendez-Gomez et al., 2024, invented a multi-lamellar RNA lipid particle aggregate (RNA-LPA) vaccine against gliomas using a lipid nanoparticle structure loaded with mRNA, which aimed to increase antigen delivery, provoking strong immune responses. In murine and canine models, intravenous administration of RNA-LPAs resulted in an increased release of cytokines, chemokines, DCs, lymphocyte trafficking and stimulation of innate immune sensors. It led to an effective rejection of both early- and late-stage tumors [191]. This study also led to a first-in-human exploratory trial in glioblastoma patients. RNA-LPA administration elicited enhanced cytokines and chemokine release, and increased the activation and trafficking of immune cells [192]. Future research on nanoparticle-based vaccines may bring about new treatments (immunotherapies) for brain tumors [192].

### 4.7. Epilepsy

In epilepsy, generalized or focal seizures result from excessive brain electrical activity. A challenge in treatment lies in the BBB’s restrictive nature, leading to insufficient anti-seizure drug concentration at the target site within the brain [193]. A nano-technological technique based on surface plasmon nanotechnology (SPN) has demonstrated potential gains in overcoming current limits in epilepsy therapy, surpassing both traditional and newly developed drug delivery methods. Promising results from recent studies have demonstrated that SPNs loaded with carbamazepine offer superior anticonvulsant effects compared to nano-emulsified carbamazepine. Similarly, muscimol SPNs and amiloride-loaded SPNs demonstrate anticonvulsant effects comparable to standalone drug administration, reducing focal seizures in rat models with enhanced and prolonged release [194]. Though the literature lacks direct reports for nanovaccines for epilepsy, this review shows that SPNs or others could be further modified for the delivery of drugs and vaccines.

### 4.8. Ischemic Stroke

An ischemic stroke is the loss of brain function that results in permanent disability due to a sudden cessation of blood and oxygen flow to the specific brain region. Hemorrhagic, lacunar, cardioembolic and cryptogenic strokes are among the various forms of vascular strokes [195]. Even though this condition carries a high rate of morbidity and mortality in the global population, there are no effective treatments to date. Moreover, brain tissue damage occurs gradually following an ischemic stroke. Hypoxia is the initial stage of an ischemic stroke, which is preceded by secondary consequences such as brain tissue inflammation, the generation of reactive oxygen species (ROS) and glutamate excitotoxicity. Brain edema, BBB disruption and nerve tissue damage gradually lead to related consequences such as neural cell death [195]. The main goals of treatment aim to provide neuroprotection while minimizing pro-inflammatory effects. The drug bioavailability across the BBB is reduced, and the current therapeutic techniques are ineffective [196]. One potential innovative treatment option to address the main obstacles in drug targeting for stroke therapy is the use of an improved nano-drug delivery system. One of the current nanotechnological approaches investigating possible medication formulations for ischemic stroke therapies is SLN carrier-based drug delivery. According to initial study results, SLNs synthesized using the high-shear homogenization approach and loaded with vincristine and temozolomide exhibit a profound, prolonged release, indicating potential clinical usage as a regulated delivery or vaccine system in the future. Regarding their use, SLNs enriched with curcumin an antioxidant have also drawn attention in stroke treatment [197].

### 4.9. Autoimmune Encephalomyelitis

The use of nanovaccination has emerged as a novel and promising approach for the treatment of neurological disorders including autoimmune encephalomyelitis, a condition characterized by the inflammation of the brain and spinal cord. Conventional therapy for autoimmune encephalomyelitis should encompass wide-ranging immunosuppressing treatment, implying severe side effects. Nonetheless, nanovaccination provides a more specific approach that can potentially fine-tune the immune system and reduce off-target effects to a minimum whilst increasing therapeutic benefit [150]. Among the greatest milestones in this area is the invention of tolerogenic nanovaccines. This is the strategy of the vaccines that are supposed to re-train the immune microenvironment to an immunosuppressive microenvironment full of regulatory T cells (Tregs). As an example, a new study has presented a core nanovaccine based on polydopamine, which is capable of triggering immune tolerance through Treg formation and expansion. The method not only inhibits the autoimmune reaction, but even prevents the destruction of the central nervous system; thus, inhibition provides a possible line of treatment for encephalomyelitis [150]. Moreover, the use of nanovaccination in autoimmune encephalomyelitis has been shown to target and regulate the antigen-presenting cells (APCs) in a specific manner, and such antigen-presenting cells are important in the development of autoimmune diseases. These nanovaccines are capable of enhancing antigen-specific tolerance, curbing inflammatory conditions and repairing damaged nervous tissues following the direct delivery of tolerogenic antigens and immunomodulatory molecules to APCs. This specific therapy method also serves the purpose of balancing the effector and regulatory immune responses, which is key to the long-term remission of the disease [150]. However, this study warrants additional future studies for confirmation of possible applications of nanovaccines or nano-based delivery systems.

### 4.10. Other Reports in Neurodegenerative Disorders

A common feature of many neurodegenerative diseases is oxidative stress, which causes neuronal cell malfunction and eventual death. Strong antioxidant support is provided by glutathione (GSH), lipoic acid (LA) and carnosine, as well as caffeic acid, in squelching the free radicals generated by ROS [198]. In one study, SLNs containing LA exhibited enhanced stability and hydrophilicity. This makes them appropriate for administering LA topically as an anti-aging agent. A report suggests that lipoyl-memantine (LA-MEM codrug)-loaded SLNs represent a novel method, enhancing absorption, stability and solubility in the gastrointestinal tract by maintaining stability in intestinal fluids and gastric fluids. This implies that at maximal concentrations, they can pass across the BBB [199]. Another popular antioxidant medication is idebenone, which is incorporated into SLNs to effectively carry the medication to the brain. Idebenone-loaded SLNs were able to suppress ROS production and 2,2′-azobis-(2-amidinopropane) dihydrochloride (APPH)-induced lactic dehydrogenase (LDH) release, according to an in vitro investigation conducted on primary cultures of rat cerebral cortex astrocytes [200]. An intriguing carrier system to cross the BBB and improve medication absorption in the brain may be provided by these particular idebenone-loaded SLNs [201]. In vitro research revealed that resveratrol, a naturally occurring polyphenolic flavonoid, and solid lipid nanoparticles loaded with grape extract could improve AD and PD associated with severe neurodegeneration and promote the regeneration of injured neurons by penetrating the BBB [202].

The Zika virus, an arbovirus, is answerable for severe neurological illnesses such as microcephaly, Guillain–Barré syndrome, meningoencephalitis and myelitis. Recently, a self-assembling nanovaccine has been created, delivering full protection against Zika virus infection. This vaccine contains the Zika virus envelope protein domain III (zEDIII) displayed on recombinant human heavy chain ferritin (rHF), generating the zEDIII-rHF nanoparticles. Mice were inoculated with the zEDIII-rHF nanoparticles without adjuvants, resulting in significant humoral and cellular responses. This immunization gave complete protection against deadly ZIKV infection and attenuated severe brain symptoms. Crucially, the immune responses induced by the zEDIII-rHF nanovaccine did not cross-react with dengue virus-2, addressing safety concerns linked to antibody-dependent enhancement (ADE) during ZIKV vaccine development. The zEDIII-rHF nanovaccine, with its exceptional protective effectiveness and prevention of ADE, appears to be a viable and safe vaccination candidate against ZIKV, which induced rare neurological disorders [203].

A multifunctional nanoplatform known as MPSDP-ZnO/Ag was developed as a hybrid ‘clusterbomb’ for the combined immunotherapy of intracranial brain tumors. High antigen loading and well-defined hybrid nano-structures were possessed by MPSDP-ZnO/Ag, and the cluster was designed to be triggered to ‘bomb’ by APCs for the simultaneous release of antigen and adjuvant. Significant antigen accumulation and enhanced cellular and humoral immunity were confirmed by the in vitro and in vivo results. Importantly, the survival time of tumor-bearing mice was evidently prolonged by nanovaccines without inducing systemic toxicity [204]. Research studies demonstrated the creation of a nanoparticle delivery system using hydrogen bonding between levodopa (L-DOPA) and tannic acid/polyvinyl alcohol (TA/PVA). This production approach resulted in nanoparticles with a small size (54 nm), a narrow size distribution and a high drug-loading capacity (46.4%). These nanoparticles displayed significant antioxidant properties when tested on an in vitro miDA neuron model subjected to H_2_O_2_-induced oxidative stress. In a rat model of Parkinson’s disease induced by reserpine, the TA/PVA/L-DOPA nanoparticles were effectively distributed in the rat brain after subcutaneous administration at the neck. This administration led to improvements in movement disorders and striatal dysfunction caused by dopamine deficiency, along with the oxidative stress associated with Parkinson’s disease. Importantly, this treatment did not result in acute toxicity to major organs. Based on these findings, it is believed that the efficient delivery of biocompatible, uniform L-DOPA-loaded nanoparticles through the brain-lymphatic vasculature holds significant potential for clinical translation in the treatment of Parkinson’s disease [205]. This provides hope for the nano-based delivery of vaccines and nanovaccines.

## 5. Next-Generation Deep Sequencing in Vaccine Delivery Systems Today and Tomorrow

In both human and microbiological genetics research, next-generation sequencing (NGS) technologies have revolutionized the field by enabling the rapid and cost-effective generation of massive sequencing datasets. This progress is now being leveraged in vaccine development, including nano-based platforms. By deciphering complex genomic and transcriptomic dynamics, NGS can guide the identification of antigens and epitopes suitable for incorporation into nanoparticle-based vaccines [206]. NGS involves several steps such as template preparation, sequencing and imaging data analysis, providing protocols that distinguish different technologies [207]. The depth of coverage, read length and chemical processes used vary among commercial platforms, while high-throughput sequencing allows for the comprehensive profiling of entire genomes or transcriptomes, facilitating systems-level analyses rather than focusing on isolated phenomena [207,208]. Key applications include genome sequencing, transcriptome profiling (RNA-Seq), DNA–protein interaction mapping (ChIP-Seq) and methylation analysis, all of which can inform vaccine design. Importantly, these NGS-derived platforms are now being employed in NP-based vaccine delivery systems, advancing the rational design of antigen-loaded carriers and nano delivery systems that may improve immune system presentation, targeting and stability. For instance, Guimaraes et al., 2024, illustrated an NP-based DNA vaccine that effectively protected animal models against SARS-CoV-2 variants, demonstrating how sequencing data could be directly translated into an optimized nanocarrier vaccine [209]. Thus, NGS-guided antigen discovery and bioinformatics analyses can facilitate the development of advanced nanovaccine platforms for cancer and neurological disorders (Figure 6).
ijms-26-10316-t001_Table 1Table 1Pre-clinical stage nano-based delivery platforms for vaccines targeting cancer and neurological disorders.Study ReferenceNanoparticles/Nanocarriers UsedSize/CompositionCirculation/Persistence/Release Profile/Delivery MechanismImmune Response Initiated and AdvantageRegulatory StatusDisorder TypeLei et al., 2024 [210]mRNA-loaded mannosylated LNPs (Man-LNPs)~100–150 nm; ionizable cationic lipids with mannose ligands for DC targeting, helper lipids (DSPC), cholesterol PEG-lipids; encapsulating uridine-modified mRNA neoantigensIM injection; mannose receptor-mediated DC uptake (3–5-fold enhanced vs. non-targeted LNPs); sustained antigen expression in 24–48 h; biodegradable, low toxicityPotent CD8^+^ T cell responses, 10-fold higher IFN-γ, Th1-biased immunity, tumor regression in E.G7-OVA lymphoma model; targeted DC activation, improved efficacy at low doses, reduced off-target effectsPre-clinicalCancer (lymphoma)Mendez-Gomez et al., 2024 [191]mRNA loaded-LNPsAggregates > 1 μm (individual LNPs ~100 nm; onion-like multilamellar structure); ionizable lipids, cholesterol, PEG-lipids; encapsulating tumor mRNA antigens/neoantigensIntravenous (IV) systemic administration; mimics emboli for lymphoreticular entrapment and RIG-I activation in stromal cells; rapid distribution (cytokine release within hours); no detailed pharmacokinetics (e.g., clearance, bioavailability) reported; well-tolerated in mice and dogsElicits rapid IFN-α/β via RIG-I/IFNAR1, enhances monocytes and lymphocytes activation, induces antigen-specific CD8^+^ T cell responses (↑ effector T cells, ↓ regulatory cells); potent immunity in glioma tumorsPre-clinical studyCancer (glioma)Chen et al., 2023 [211]Acid-ionizable iron nano-adjuvant (PEIM; IONPs with STING agonist)~50–100 nm; iron oxide nanoparticles (IONPs) coated with acid-ionizable copolymers like PEI, co-assembled with STING agonist MSA-2; encapsulating personalized tumor antigensIntratumoral injection; concentrates in draining lymph nodes; acid-responsive release of Fe^3+^ for ROS generation and STING activation; facilitates APCs uptakeAugments STING/IFN-I pathway, enhances antigen cross-presentation, elicits 55-fold higher CD8^+^ T cell response; advantages: 16-fold STING agonist dosage-sparing, potent anti-tumor immunity in melanoma and colorectal carcinoma modelsPre-clinicalCancer (melanoma and colorectal carcinoma)Pan et al., 2023 [212]Stearic acid-doped LNPs (sLNPs-OVA/MPLA)~100–150 nm; ionizable cationic lipid, DSPC, cholesterol, DMG-PEG2000, stearic acid (anionic); co-loaded with OVA-mRNA and MPLA (TLR4 agonist)IV injection; spleen-selective mRNA translation via stearic acid; sustained antigen expression in 24–48 hEnhanced DC activation, Th1-biased CD8^+^ T cell responses (↑ IFN-γ), persistent immune memory; potent tumor growth inhibition in E.G7-OVA lymphoma and B16F10-OVA melanoma models; synergistic TLR4 activationPre-clinicalCancer (lymphoma and melanoma)Cao et al., 2023 [61]Dendrimers; GT-Mn^2+^ coordinative dendrimers~100–200 nm; amine-terminated PAMAM dendrimers (G5) coordinated with Mn^2+^ ions for self-assembly; encapsulates peptide antigensSubcutaneous (SC) injection; DC internalization via macropinocytosis/lipid-raft pathways; gradual dissociation for antigen release and Mn^2+^ activationEfficient antigen cross-presentation on MHC-I, activates cGAS-STING pathway (↑ IFN-β, cytokines); induces robust CD8^+^ T cell responses, Th1 immunity; advantages: personalized neoantigen packaging, potent tumor regression in melanoma or lymphoma modelsPre-clinicalCancer (lymphoma and melanoma)Shen et al., 2023 [62]Dendrimers; photothermal-triggered dendrimers (IR780-PAMAM-OVA)~100–150 nm; PAMAM dendrimer (G5) conjugated with IR780 (photothermal agent) and OVA peptide antigen; self-assembled nanoparticlesSC injection; photothermal-triggered antigen release under NIR (near infrared) laser (808 nm); lymph node drainage; sustained release post-NIR-exposure; well-toleratedActivates DCs via photothermal effect, enhances antigen-specific CD8^+^ T cell responses, Th1 immunity (↑ IFN-γ, IL-12); NIR-controlled release boosts immunity, significant tumor suppression in B16-OVA melanoma modelPre-clinicalCancer (melanoma)Sasaki et al., 2022 [213]mRNA-loaded LNPs (A-11-LNP; DC-targeted)~200 nm (optimal range; A-11: 547 nm); pH-sensitive cationic lipid CL4H6 (60%), DOPE (10%), cholesterol (28.5%), PEG-DSG (1.5%); encapsulating mRNA containing antigensIV administration; targets splenic DCs; higher uptake/gene expression vs. smaller LNPs; transgene expression peaks at 24 h; multi-dosing toleratedInduces DC-specific transgene expression (↑ CD40/CD80/CD86), antigen-specific CD8^+^ T cell responses; superior anti-tumor efficacy in E.G7-OVA lymphoma model; low toxicityPre-clinicalCancer (lymphoma)Kozaka et al., 2019 [65]Micelles (reverse micellar antigen carriers)~10–20 nm; reverse micelles with sucrose erucate (ER-290), cholesterol and phosphatidylcholine; encapsulates OVA protein or tumor antigensIntradermal; reverse micelles penetrate stratum corneum, target cutaneous DCs; sustained antigen release; well-toleratedInduces antigen-specific CD8^+^ T cell responses, Th1-biased immunity (↑ IFN-γ); non-invasive transcutaneous delivery, effective tumor suppression in melanoma model, simpler than invasive methodsPre-clinicalCancer (melanoma)Kranz et al., 2016 [214]Liposomes/LNPs/RNA-lipoplex (RNA-LPX)~200–400 nm; ionizable cationic lipids, tumor antigen mRNAIV delivery; targets spleen DCs via net negative charge; rapid uptake (within hours) into lymphoid DCs or macrophages; measurable half-life in circulation; persists in lymphoid organs; protects RNA from degradationInduces IFN-α release, antigen-specific CD4^+^/CD8^+^ effector/memory T cell responses; IFNα-dependent tumor rejection in B16-OVA melanoma mouse models; ligand-free DC targeting, potent innate or adaptive immunity, broad antigen applicabilityPre-clinicalCancer (melanoma)Lee et al., 2023 [154]Dendrimers~100–150 nm; polyamidoamine (PAMAM) dendrimers conjugated with amyloid-beta (Aβ) peptide (1–42)SC injection; lymph node drainage for DC uptake; sustained antigen presentation; well-toleratedInduces anti-Aβ antibodies (IgG) and Aβ-specific regulatory T cells (Tregs); reduces Aβ plaques, neuroinflammation and cognitive deficits in APP/PS1 mouse model of Alzheimer’s; balances humoral immunity and immune regulation to avoid excessive inflammationPre-clinicalNeurological disorder (Alzheimer’s)Gomi et al. [169]PS-LNPs (phosphatidylserine lipid nanoparticles)~131–133 nm; negative zeta potential ≈ −21 mV; encapsulating MOG_35–55_ mRNA antigenIV injection; spleen-targeting; antigen presentation of self-antigen; low dose (1 µg) temporal dosing on days 7, 10, and 13 post-immunization; sustained release, well toleratedInduces antigen-specific tolerance; lowers EAE (experimental autoimmune encephalomyelitis clinical score) scores, reduces IL-17A and pro-inflammatory cytokines; Treg induction; antigen specificity and low dosePre-clinicalNeurological disorder (multiple sclerosis)Tables Legends: ↑ shows an upregulation, ↓ shows a downregulation of markers, and all other abbreviations have been explained in the text.
ijms-26-10316-t002_Table 2Table 2Clinical stage nano-based delivery platforms for vaccines targeting cancer and neurological disorders.Study ReferenceNanoparticles/Nanocarriers UsedSize/CompositionCirculation/Persistence/Release Profile/Delivery MechanismImmune Response Initiated and AdvantageRegulatory StatusDisorder TypeWeber et al., 2024 [215]LNP formulation, personalized neoantigen lipid nanoparticles~80–100 nm; lipid nanoparticle (LNP) formulation with ionizable cationic lipids, cholesterol, PEG-lipids; synthetic mRNA encoding up to 34 patient-specific neoantigensIntramuscular (IM) injection; designed for lymph node drainage and DC uptake; sustained antigen expression in days; well-toleratedInduces neoantigen-specific CD4^+^/CD8^+^ T cell responses; individualized for melanoma tumor mutationsPhase II clinical trial completed (2024, positive); multiple Phase III ongoing; not FDA approved yet; patentedCancer (melanoma)Mendez-Gomez et al., 2024 [191]RNA lipid particle aggregates (RNA-LPAs; multi-lamellar mRNA aggregates)Aggregates > 1 μm (individual LNPs 100 nm, onion-like multilamellar); composed of ionizable lipids, cholesterol, PEG-lipids; encapsulating patient-derived tumor mRNA antigens like IL13Rα2Intravenous (IV) systemic administration; rapid systemic distribution; well-tolerated with multi-dosingStimulated rapid cytokines, chemokines release, monocytes and lymphocytes activation and antigen-specific CD8^+^ T cell expansion, reduced regulatory cells, increased effector T cells; enhanced immunogenicity for glioblastomaPhase I clinical trial completed; safe; not FDA approved; not patented yetCancer (glioblastoma)Rojas et al., 2023 [216]RNA-lipoplex nanoparticles (iNeST; autogene cevumeran RO7198457/BNT12)~200–400 nm; proprietary ionizable cationic lipids, patient-derived tumor neoantigen mRNA (up to 20 neoantigens)IV delivery; targets splenic DCs via net negative charge; rapid uptake in hours; sustained T cell persistence up to 3 years; well-tolerated in multi-dosingInduces neoantigen-specific CD8^+^ T cells responses detected up to 3 years; fully personalized for tumor mutations, enhances anti-tumor immunity in pancreatic and melanoma tumorsPhase I clinical trial completed; multiple Phase II trials ongoing; not FDA approved; not patented yetCancer (melanoma and pancreatic cancer)Sahin et al., 2020 [217]Liposomes; RNA-lipoplex (RNA-LPX; FixVac BNT111)200–400 nm; proprietary ionizable cationic lipids, uridine-modified mRNA encoding four melanoma antigens (NY-ESO-1, MAGE-A3, tyrosinase, TPTE)IV delivery; targets splenic dendritic cells via net negative charge; rapid uptake in hours; sustained antigen expression; well-tolerated up to 400 µg doses in multi-dosing regimensInduces IFN-α, durable antigen-specific CD4^+^/CD8^+^ T cell responses in melanoma, enhances pre-existing immunityPhase I clinical trial completed (2020); Phase 2 ongoing (2024 topline positive); not FDA approved; not patented yetCancer (melanoma)T. Gargett et al., 2018 [30]Liposomes/LNPs (dendritic-cell-targeted nanocarriers)~100–150 nm; MM200 melanoma cell vesicles, POPC liposomes, anti-DC-SIGN antibodies, IFN-γIV delivery; targets DC-SIGN on DCs for antigen presentation; multi-dose toleratedInduced antigen-specific T cell responses (CD4^+^ and CD8^+^) and antibody production; strategy designed to enhance presentation and overcome immune tolerance in melanomaPhase I clinical trial completed; safe; not FDA approved; patentedCancer (melanoma)Kranz et al., 2016 [214]Liposomes/LNPs, RNA-lipoplex (RNA-LPX)~200–400 nm; ionizable cationic lipids, tumor antigen mRNAIV delivery; targets spleen DCs via negative charge; well-toleratedInduces IFN-α, antigen-specific CD4^+^/CD8^+^ T cell responses; systemic DC targeting, potent T cell priming for melanomaPhase I clinical trial completed; safe; not FDA approved; not patented yetCancer (melanoma)Palmer et al., 2001 [218]Liposome/LNPs~100–200 nm (multilamellar); BLP25 lipopeptide (25-aa MUC1 core peptide), MPL adjuvant, DPPC, DMPG, cholesterolIntradermal; prolonged circulation (PEGylated)/controlled antigen/drug release, cyclophosphamide pretreatment; facilitates APC uptake for MHC presentation; well-tolerated multi-dosingInduces MUC1-specific T cell (IFN-γ, proliferation) and humoral responses; safe with minimal toxicity; targets overexpressed MUC1 in NSCLC (non-small-cell lung cancer) for active specific immunotherapyPhase I and II clinical trials completed; advanced to Phase III; not approved by FDA; patentedCancer (non-small-cell lung cancer)Ciccone, 2024[159]B-312 nano synthetic peptides/antigensSynthetic peptides derived from α-synucleinIM injection; doses tested: 300/100/100 µg and 300/300/300 µg; antibody titers peaked around week 21, detectable up to ~45 weeks; antibodies cross blood–brain barrierInduces antibodies selectively against aggregated α-synuclein; reduces pathological α-synuclein seeding activity in CSFPhase I clinical trial completed; not approved by FDA; not patented yetNeurological disorder (Parkinson’s)Eijsvogel et al., 2024 [160]LNPs~80–100 nm; ionizable lipid, DSPC, cholesterol, PEG-lipid; peptide delivery as antigensIntramuscular injection; induces anti-α-synuclein antibodies; well tolerated; crossed BBBInduces antibodies selectively against aggregated α-synuclein; reduces pathological α-synuclein seeding activity in CSF, improves cognitive behaviorPhase II clinical trial completed; not approved by FDA; not patented yetNeurological disorder (Parkinson’s)Volc et al., 2020 [161]Active peptides/nano antigensSynthetic peptides derived from α-synucleinIM injection (300/100/100 μg or 300/300/300 μg doses, 3 doses over 8 weeks); systemic distribution for immune activation; antibodies detectable in serum and CSF up to 44 weeksInduces anti-α-Syn antibodies; reduces pathological α-Syn seeding in CSF, improves MDS-UPDRS Part II scores (daily living activities), stable motor/cognitive function; potentially disease-modifying for early PDPhase I clinical trial completed but not approved by FDA; not patented yetNeurological disorder (Parkinson’s)Yu et al., 2023[219]Peptides (synthetic Aβ_1–14_ B cell epitope peptides linked to UBITh^®^ helper T cell epitopes; alum + CpG adjuvants)Synthetic micro/nano peptidesIntramuscular injection; distribution for immune activation; antibodies detectable in serum and CSFInduces robust anti-amyloid β antibodies; high responder rate; has shown trends toward slowing cognitive decline; favorable safety profilePhase II clinical trial completed; not approved by FDA; patentedNeurological disorder (Alzheimer’s)


## 6. Future Challenges and Perspectives

The future of nano-based vaccine delivery systems and nanovaccines holds immense promise, but also considerable challenges. Nanovaccines have the potential to revolutionize preventive medicine by offering targeted and efficient delivery of antigens, enhancing immune responses and enabling the development of novel vaccine formulations. However, several significant hurdles must be addressed to realize their full potential. One major challenge is ensuring the safety and biocompatibility of nanovaccine carriers to minimize adverse reactions. Additionally, optimizing the design and manufacturing processes to scale up production while maintaining consistency and quality is crucial for widespread adoption. Furthermore, there is a need for comprehensive understanding of the immune responses elicited by nanovaccines to tailor their design for specific populations and disorders. Regulatory approval and public acceptance are also key considerations in the implementation of nanovaccine delivery systems. Despite these challenges, continued research and collaboration among scientists, clinicians, regulators and industry stakeholders offer a pathway toward harnessing the transformative power of nano-based vaccine delivery systems and nanovaccines for global health.

## 7. Conclusions

Vaccine-nanoparticle formulation benefits greatly from improved Ag cellular uptake and potential targeting to APCs with an improved immunogenicity stemming from either co-adjuvant molecules conveyed by the NPs or immune-stimulatory properties of the NPs. Most vaccine-nanoparticle systems being currently considered are biocompatible and biodegradable with minimum toxicity, offering effective and safe alternatives to traditional vaccines. General mechanisms of nanoparticle-based vaccine delivery systems or nanovaccines that can upgrade vaccines may include targeting, the stimulation of immune responses and cellular uptake of vaccines, and the efficacy and penetrability of drugs crossing biological barriers. A wide range of unique nanoparticle-based formulations differing in size, composition, route of administration and charge exhibit crucial roles in the modulation of biodistribution, overall immune responses, and cellular trafficking. The nano-based technology field is primed to address current challenges in immunology and foster the transformation of ground-breaking strategies in the future of vaccine development and design.

## Figures and Tables

**Figure 1 ijms-26-10316-f001:**
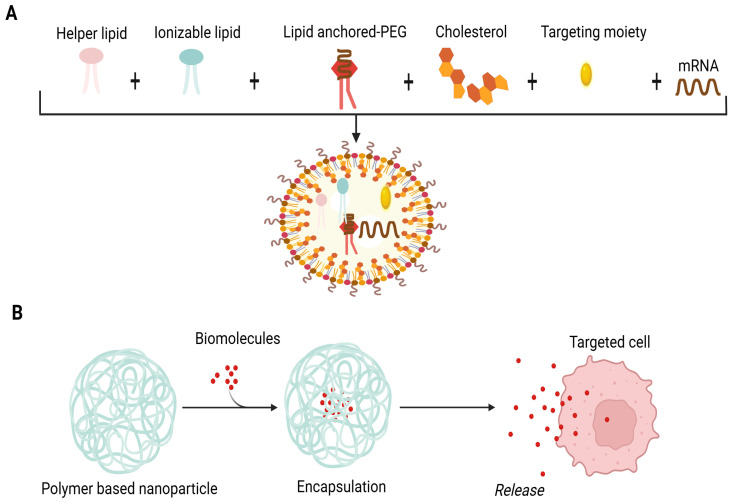
Composition of LNPs. (**A**) Lipid NPs are composed of four lipid components: helper lipid for cargo encapsulation, ionizable lipid for enhancement of endosomal escape and delivery, cholesterol for stability preferment, and lipid-affixed polyethylene glycol (PEG) for reduction of immune system recognition and improvement of biodistribution. Antibodies or peptides like directing moieties may be added to direct localization. Lipid components are pooled with mRNA through microfluidic mixing to form lipid NPs. (**B**) Schematic illustration of biomolecules’ encapsulation in the polymer matrix. The cross-linkage of the polymer matrix allows the packing of biomolecules and expedites their release by degradation of the matrix.

**Figure 2 ijms-26-10316-f002:**
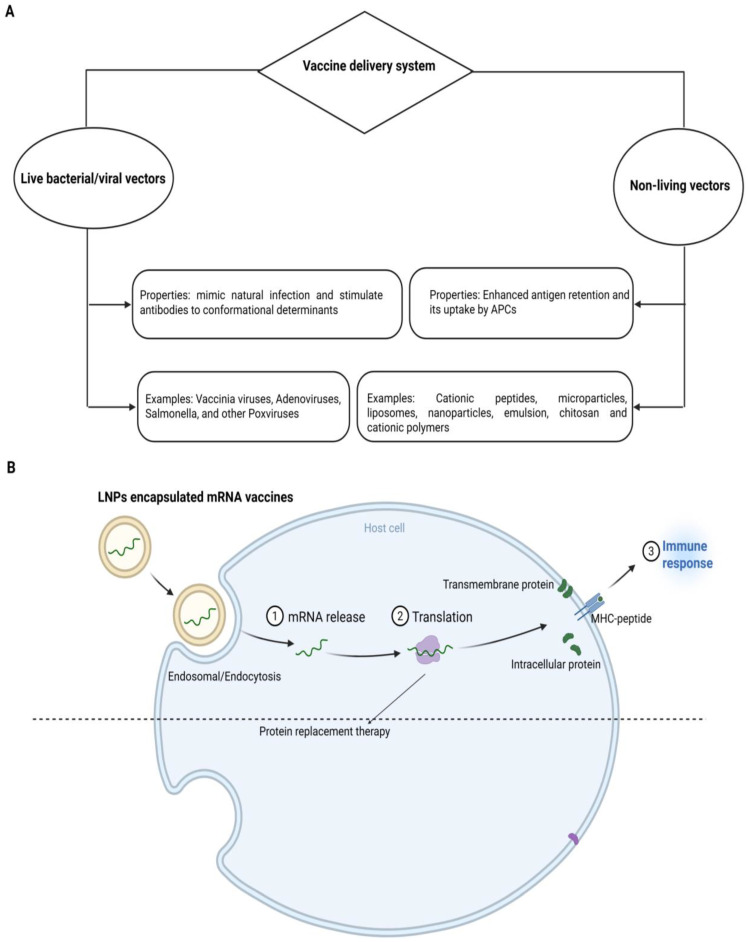
Systems of vaccine delivery. (**A**) Live or biologically-based gene delivery systems and non-live delivery systems. Non-live delivery systems include liposomes, cationic lipids, polysaccharides, cationic peptides, micro/NPs and cell-penetrating peptides. (**B**) Mode of operation of mRNA-based vaccines. As nucleases degrade the mRNA and this degraded mRNA cannot cross the cell membrane owing to its negative charge and large size, delivery demands encapsulation in vehicles, e.g., lipid NPs. Cellular uptake of lipid NPs starts with the endocytosis process followed by the endosomal body escape, degradation of lipid NPs and release of mRNA into the cytosol. For therapeutic applications, mRNA is translated into protein to act as vaccines or protein replacement therapy.

**Figure 3 ijms-26-10316-f003:**
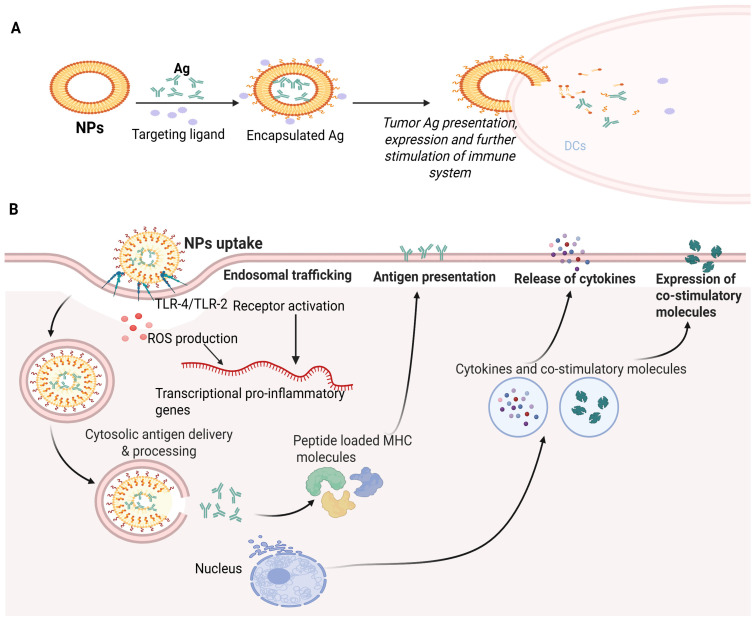
Immunotherapy based on NPs. (**A**) Outline of the mechanism of immunotherapy based on an NP-dependent tumor peptide vaccine. (**B**) Molecular mechanisms for immune stimulation through NP-based vaccination. The NPs can have immuno-stimulating properties upon uptake through receptor (TLR-4) activation and ROS. These pathways transcribe pro-inflammatory genes, resulting in the translation of co-stimulatory molecules and pro-inflammatory cytokines. NP-based cytosolic delivery of Ags ensures that the Ag enters Ag processing. Peptidase and proteasome-mediated machinery deliver oligopeptides (processed Ags) to be presented in MHC molecules which are exposed to the cell’s surface. Increased cytosolic Ag delivery with immune-stimulatory properties of NPs results in effective priming of Ag-specific T cells.

**Figure 4 ijms-26-10316-f004:**
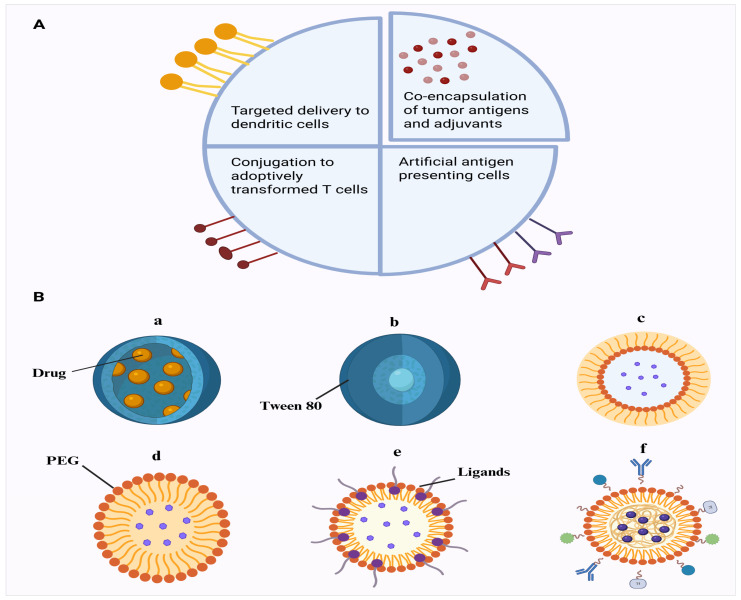
Multi-functional design of nanoparticles for cancer therapy. (**A**) The nanoparticle surface can be functionalized with antigens (Ags) or antibodies specific to dendritic cells (DCs), while adjuvants and tumor antigens can be co-loaded into the nanoparticle core. Additionally, MHC–antigen complexes and co-stimulatory binding ligands can be incorporated to act as synthetic APCs. Additionally, NPs with immune potentiators can be conjugated on T cells to ameliorate adoptive T cell therapy. (**B**) An example of polymeric nanoparticles used in neurodegenerative diseases treatment. (**a**) Nanocapsules comprise core shell structured nanoparticles in which the drug is enclosed within a polymeric membrane. (**b**) Tween 80-coated PBCA nanoparticles compose poly(butyl cyanoacrylate) particles coated with surfactant Tween 80 to facilitate blood–brain barrier penetration. (**c**) Pegylated nanospheres encompass polymeric nanospheres surface-modified with PEG to improve circulation time and biocompatibility. (**d**) Nanospheres coated with ligands and/or antibodies constitute functionalized particles for targeted delivery to specific receptors. (**e**) Nanospheres coated with ligands and/or antibodies. (**f**) Pegylated nanospheres comprise extra ligands and antibodies as dual-modified nanospheres combining stealth (PEGylation) and receptor-specific targeting. This mechanism provides hope for development of nanoparticle-based vaccines in future.

**Figure 5 ijms-26-10316-f005:**
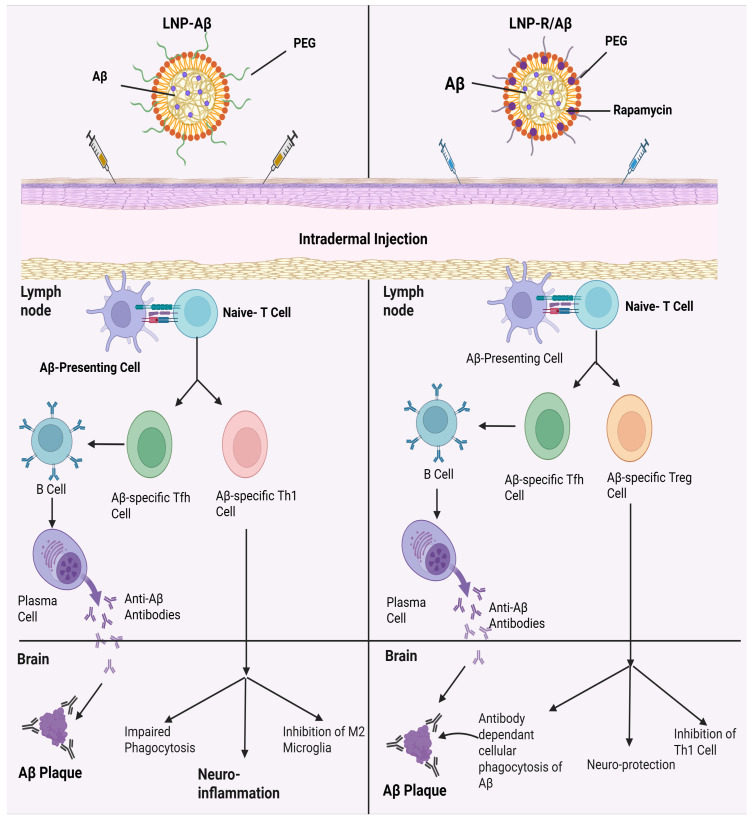
The mechanism of action and adverse effects of LNP-Aβ (neuroinflammation) as well as suggested therapeutic strategies using LNP-R/Aβ. The LNP-Aβ vaccination can produce anti-Aβ antibodies; however, it also has meningitis side effects. On the other hand, LNP-R/Aβ exhibits neuroprotective properties and can eliminate Aβ plaques without causing neuroinflammation. When the LNP-Aβ vaccine is injected intradermally, Aβ-specific Th1 and Tfh cells are produced, which eliminates Aβ plaque and causes neuroinflammation. By contrast, the production of Aβ-specific Tfh cells and Treg cells following intradermal injection of LNP-R/Aβ results in the elimination of Aβ plaque and neuroprotection.

**Figure 6 ijms-26-10316-f006:**
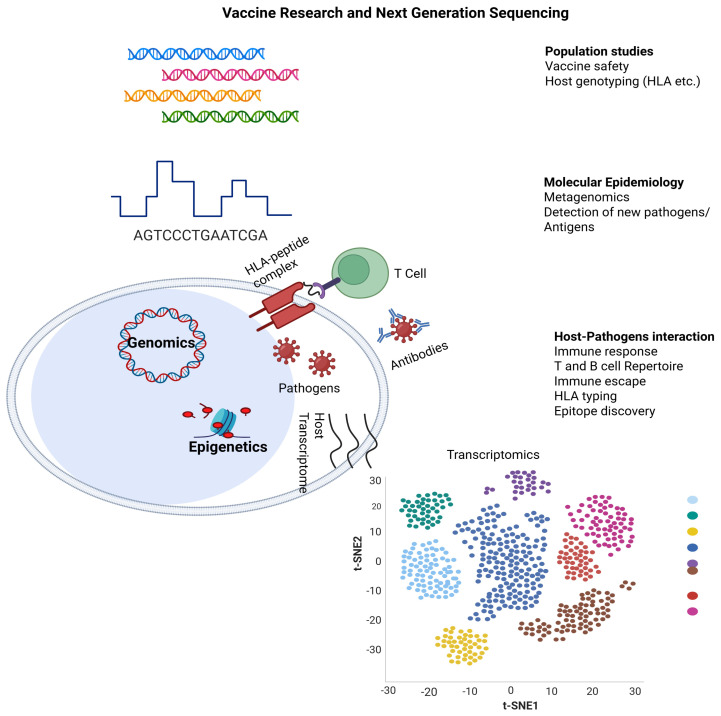
Next-generation sequencing (NGS) and its role in vaccine development. The figure illustrates how NGS influences multiple aspects of vaccine research. NGS is employed in longitudinal studies of host–pathogen interactions, transcriptome assessment and immune response analyses such as profiling T and B cell repertoire diversity. It also aids in identifying novel antigens, evaluating vaccine stockpiles and assessing HLA polymorphism diversity across populations. The right-bottom panel presents a t-distributed stochastic neighbor embedding (t-SNE) plot illustrating single cell transcriptomic clustering based on gene expression profiles, where each colored dot corresponds to a distinct immune cell subtype. The t-SNE1 (x-axis) and t-SNE2 (y-axis) represent two dimensions of reduced gene expression data, with each cluster reflecting transcriptional similarity among immune cells. This visualization demonstrates how NGS data can be used to characterize immune heterogeneity and vaccine-induced cellular responses. Overall, the figure highlights the comprehensive role of NGS in advancing vaccine development, including applications in nanovaccines for cancer and neurological disorders.

## Data Availability

Datasets generated during and/or analyzed for this study have been included in the main text.

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
