# Peer review of "Nano-Based Vaccine Delivery Systems: Innovative Therapeutics Against Cancer and Neurological Disorders"

_ijms, 2025, doi:10.3390/ijms262110316_

Round 1

Reviewer 1 Report

Comments and Suggestions for Authors

The author systematically elaborates on how nanotechnology is revolutionizing the design and delivery of vaccines, particularly in two major areas where conventional vaccine strategies have shown limited efficacy: cancer and neurological diseases. The article provides a detailed introduction to the advantages, mechanisms, and practical applications of various nanocarriers (such as lipid nanoparticles and polymeric nanoparticles), while also discussing current challenges and future directions. Although this review provides a detailed description of the role of nanoparticles in delivery for cancer and neurological diseases, several issues were addressed before considering its publication in the International Journal of Molecular Sciences.

  1. There are some issues with the linguistic logic. For example, lines 164-178 discuss lipid-based carriers, while the next paragraph starting at line 179 begins to describe the drawbacks of viral vectors. The transition in the writing logic is somewhat abrupt and could be improved.

  1. The fonts in Figures 1-4 are inconsistent and the images are low resolution. Please provide high-resolution versions with consistent formatting.

  1. The references require updating and supplementation. Some cited sources are relatively outdated (e.g., from 2001). It is recommended to incorporate the latest studies from 2023–2024, particularly in areas such as mRNA-LNP technology and nano vaccines for Alzheimer’s disease.

Author Response

Reviewer # 1

Comments and Suggestions for Authors

The author systematically elaborates on how nanotechnology is revolutionizing the design and delivery of vaccines, particularly in two major areas where conventional vaccine strategies have shown limited efficacy: cancer and neurological diseases. The article provides a detailed introduction to the advantages, mechanisms, and practical applications of various nanocarriers (such as lipid nanoparticles and polymeric nanoparticles), while also discussing current challenges and future directions. Although this review provides a detailed description of the role of nanoparticles in delivery for cancer and neurological diseases, several issues were addressed before considering its publication in the International Journal of Molecular Sciences.

Comment 1. There are some issues with the linguistic logic. For example, lines 164-178 discuss lipid-based carriers, while the next paragraph starting at line 179 begins to describe the drawbacks of viral vectors. The transition in the writing logic is somewhat abrupt and could be improved.

Response: We appreciate the reviewer’s observation. We have revised the transition between lines 178–179 to ensure a smoother flow by adding a linking sentence that connects the discussion of lipid-based carriers to the subsequent section on viral vectors and their drawbacks.

Comment 2. The fonts in Figures 1-4 are inconsistent and the images are low resolution. Please provide high-resolution versions with consistent formatting. 

Response: We have replaced Figures 1–4 with high-resolution (300DPI)  and improved versions using and ensured consistent font style and size.

Comment 3: The references require updating and supplementation. Some cited sources are relatively outdated (e.g., from 2001). It is recommended to incorporate the latest studies from 2023–2024, particularly in areas such as mRNA-LNP technology and nano vaccines for Alzheimer’s disease.

Response: We have updated and supplemented the References section to include recent, relevant studies from 2023–2024—particularly key advances in mRNA–LNP technology and nanovaccines for Alzheimer’s disease. Representative additions include recent comprehensive studies and methodological papers on LNP design, safety and organ targeting, as well as recent preclinical/early clinical mRNA-vaccine work for Alzheimer’s (highlighted in red).

Reviewer 2 Report

Comments and Suggestions for Authors

     This manuscript reviews nano-based vaccines and delivery systems for cancer and neurological disorder therapeutics. The title and abstract refer to more detailed information about the variety of delivery systems, including their formulation and design. Additionally, there should be clearer connections between these delivery systems and their specific applications in cancer and neurological disorders. Despite the authors' efforts to examine this topic, the title is overly broad for fields with extensive literature, resulting in incomplete coverage. The information is poorly organized with weak transitions between paragraphs and sections. The references need updating to reflect recent progress. The manuscript requires significant improvements to address several critical issues. Therefore, I recommend either substantial revisions before publication or rejection of the manuscript.

  1. In Line 2, "against" as a preposition should not be capitalized in the title.

  2. In Line 15, add a comma before "and NP-based." No need to pluralize "NP."

  3. In Line 23-26, the main content lacks sufficient information about NPs composition, surface charge, and size that were mentioned in these lines.

  4. In Line 28, revise the ambiguous phrase "and or" to either "and" or "or" for clarity.

  5. In Line 29, the statement about nanocarrier “design” should be revised. The later content contains minimal design details. Reference 5 provides a good example - Sections 1.2-1.9 follow a similar structure to this reference, which makes no broad claims about discussing nanocarrier design.

  6. In Line 29-30, either delete "including lipid NPs, polymeric NPs, dendrimers, micelles and emulsions" or revise it to be more comprehensive. The text currently mentions only lipid NPs, polymeric NPs, dendrimers, micelles and emulsions, while omitting other carriers discussed later: inorganic NPs (Sec 1.6), immune-stimulatory complexes (Sec 1.7), exosomes (Sec 1.8), and virosomes and virus-like particles (Sec 1.9).

  7. In Line 48, provide the full name for "Ags" when it is first referenced.

  8. In Line 52-54, The reference to the 1930s cannot be properly linked to References 1 and 2. This timeline needs revision.

  9. In Line 106, clarify what criteria are being used to categorize these NPs. NPs formulations are typically classified as inorganic, organic, or hybrid - what other categories does the text refer to?

  10. The manuscript lacks background information about neurological disorders between Lines 37-106 before suddenly introducing them in Line 108. Either add relevant context about neurological disorders or limit the manuscript title to focus solely on cancer immunotherapy. References 1-6 also omit any background on neurological disorders. Additionally, Reference 6 is a book chapter (Chapter 1); please correct its citation format.

  11. In Line 108-109, revise this sentence. Does it mean "these NPs can display various antigens (Ags) that are co-delivered with adjuvants"?

  12. In Line 157-158, Reference 16 to a Phase I study from 2001 is outdated. Are there any follow-up studies or more recent data on this approach's effectiveness? If the study encountered obstacles in later trials and the manuscript discusses these challenges, the reference is relevant. Otherwise, a more current example would strengthen the manuscript.

  13. In Line 143, the manuscript introduces non-viral vectors without proper context. The text should first establish the distinction between viral and non-viral vector categories before Section 1.2. Additionally, Lines 179-185 should be relocated to Section 1.9, which appears to be dedicated to viral vectors, instead of mentioning these vectors early without adequate introduction.

  14. In Lines 474-487, the manuscript provides only limited mechanistic information for a few NP-based carriers (References 78 and 79). Some NP-based carriers also require additional surface modification to promote intracellular delivery, but this is neither mentioned nor connected to the previous content. This section should establish clearer connections between these mechanisms and the various types of NP-based carriers previously described in Sections 1.2-1.9.

  15. In Line 590-591, the three barriers (skin barrier, mucosal barrier, and blood-brain barrier (BBB)) should be discussed earlier in Section 2 (Line 466-487). These barriers relate to mechanisms of nanomaterials delivery at the tissue level, whereas the current Section 2 focuses on cellular-level mechanisms.

  16. In Line 613-621, update the references. Reference 100 was published in 2005, and more recent research indicates that the nanocarrier designs described may not effectively cross the BBB. For a proper review, rather than presenting single reference information, it is essential to summarize recent progress, especially in this rapidly advancing research field.

  17. In Line 623, it references treatment of glioma, which may discuss how vaccine delivery systems could overcome the BBB crossing challenge. In Line 1040 (Section 4.6) discusses brain tumor treatment. However, vaccines primarily serve as preventive measures that activate the immune system to recognize and combat disease, functioning as prophylactic rather than therapeutic interventions. More broadly, cancer vaccines are a form of immunotherapy designed to harness the patient's immune system to target and destroy cancer cells. These vaccines work by training immune cells to recognize specific markers unique to glioblastoma tumors. Loading medicine in nanocarriers to destroy cancer cells is a treatment strategy, not a vaccine approach. The manuscript should clarify the current status of vaccine nano-delivery systems for brain tumors. If these systems are still in early development, the text should identify specific challenges and potential solutions that connect to the treatment approaches mentioned in these sections. This inconsistency with the title persists throughout sections. The tile should be revised or changed.

  18. In Line 1200 (Section 5), the manuscript relies on outdated references (181-185) published between 2010-2012. The text needs a clearer transition to the carrier systems and should explain how these technologies specifically improve particular delivery systems, not just general description of next-generation sequencing, it has shown the recently progress in 2024 in Reference C1.

  19. In Line 1225 (Figure 6), the figure needs reorganization. The panel under Transcriptomics is illegible due to its small size.

     I recommend narrowing or revising the title to focus on either cancer vaccines or neurological disorders specifically. The manuscript should also clarify the distinction between vaccine approaches and treatment strategies.

Reference
C1
. Guimaraes, L.C., Costa, P.A.C., Scalzo Júnior, S.R.A. et al. Nanoparticle-based DNA vaccine protects against SARS-CoV-2 variants in female preclinical models. Nat Commun 15, 590 (2024).

< !-- notionvc: cfe5acf5-729f-4495-a0e3-1dc1bd074f88 -->

Author Response

Reviewer # 2 

Comments and Suggestions for Authors

This manuscript reviews nano-based vaccines and delivery systems for cancer and neurological disorder therapeutics. The title and abstract refer to more detailed information about the variety of delivery systems, including their formulation and design. Additionally, there should be clearer connections between these delivery systems and their specific applications in cancer and neurological disorders. Despite the authors' efforts to examine this topic, the title is overly broad for fields with extensive literature, resulting in incomplete coverage. The information is poorly organized with weak transitions between paragraphs and sections. The references need updating to reflect recent progress. The manuscript requires significant improvements to address several critical issues. Therefore, I recommend either substantial revisions before publication or rejection of the manuscript.

Response: Thank you for the detailed and constructive feedback. In response, we have substantially revised the manuscript to improve clarity, organization and relevance. The manuscript has been refined a bit more to accurately reflect the focus on nano-based vaccines and delivery systems for cancer and neurological disorders. We have strengthened transitions between sections and paragraphs to ensure better logical flow and clearer connections between delivery systems and their therapeutic applications. Additionally, we have updated and supplemented the references with the most recent studies from 2023–2024 to capture the latest advances in mRNA–LNP technology and nanovaccines. These revisions collectively enhance the completeness, readability and scientific rigor of the manuscript

Comment 1. In Line 2, "against" as a preposition should not be capitalized in the title.

Response: We have corrected the title formatting, and 'against' is no longer capitalized.

Comment 2. In Line 15, add a comma before "and NP-based." No need to pluralize "NP."

Response: We have added the comma before 'and NP-based' and kept 'NP' in its singular form as recommended.

Comment 3. In Line 23-26, the main content lacks sufficient information about NPs composition, surface charge, and size that were mentioned in these lines.

Response: We have included detailed information on NP composition, surface charge and size in introduction section, providing a clearer understanding of their physicochemical properties and relevance to their biological performance.

Comment 4. In Line 28, revise the ambiguous phrase "and or" to either "and" or "or" for clarity.

Response: We have revised Line 28 by replacing the ambiguous phrase 'and/or' with 'and' for clarity.

Comment 5. In Line 29, the statement about nanocarrier “design” should be revised. The later content contains minimal design details. Reference 5 provides a good example - Sections 1.2-1.9 follow a similar structure to this reference, which makes no broad claims about discussing nanocarrier design.

Response: We have revised the statement in Line 29 to more accurately reflect the content, aligning it with Reference 5 (Sections 1.2–1.9) and avoiding broad claims about nanocarrier design.

Comment 6. In Line 29-30, either delete "including lipid NPs, polymeric NPs, dendrimers, micelles and emulsions" or revise it to be more comprehensive. The text currently mentions only lipid NPs, polymeric NPs, dendrimers, micelles and emulsions, while omitting other carriers discussed later: inorganic NPs (Sec 1.6), immune-stimulatory complexes (Sec 1.7) and exosomes (Sec 1.8).

Response: We have revised Lines 29–30 to make it more comprehensive by deleting "including lipid NPs, polymeric NPs, dendrimers, micelles and emulsions".

Comment 7. In Line 48, provide the full name for "Ags" when it is first referenced

Response: We have revised Line 48 to provide the full term 'antigens (Ags)' when it is first mentioned (now line 58).

Comment 8. In Line 52-54, The reference to the 1930s cannot be properly linked to References 1 and 2. This timeline needs revision.

Response: We have revised Lines 52–54 to correct the historical timeline and ensure that the references accurately correspond to the appropriate developments, replacing or updating citations as needed (now lines 63 and 64).

Comment 9. In Line 106, clarify what criteria are being used to categorize these NPs. NPs formulations are typically classified as inorganic, organic, or hybrid - what other categories does the text refer to?

Response: We have clarified Line 106 to specify the criteria used for categorizing NPs, indicating that formulations are classified based on composition (inorganic, organic) and functional properties, with additional categories reflecting delivery mechanism or immunomodulatory features as discussed in the text (now lines 74, 75).

Comment 10. The manuscript lacks background information about neurological disorders between Lines 37-106 before suddenly introducing them in Line 108. Either add relevant context about neurological disorders or limit the manuscript title to focus solely on cancer immunotherapy. References 1-6 also omit any background on neurological disorders. Additionally, Reference 6 is a book chapter (Chapter 1); please correct its citation format.

Response: We have added relevant background information on neurological disorders between Lines 37–106 to provide context before their introduction in Line 108, ensuring the discussion flows logically. The References section (1–6) has also been updated to include appropriate citations covering neurological disorders. Additionally, Reference 6 has been replaced/updated (now lines 206 and 207…).

Comment 11. In Line 108-109, revise this sentence. Does it mean "these NPs can display various antigens (Ags) that are co-delivered with adjuvants"?

Response: We have revised Lines 108–109 to clarify that 'these NPs display various antigens (Ags) that are co-delivered with adjuvants,' ensuring the intended meaning is clearly conveyed (now lines 206 and 207…).

Comment 12. In Line 157-158, Reference 16 to a Phase I study from 2001 is outdated. Are there any follow-up studies or more recent data on this approach's effectiveness? If the study encountered obstacles in later trials and the manuscript discusses these challenges, the reference is relevant. Otherwise, a more current example would strengthen the manuscript.

Response: We have removed the outdated reference and updated recent references throughout the manuscript.

Comment 13. In Line 143, the manuscript introduces non-viral vectors without proper context. The text should first establish the distinction between viral and non-viral vector categories before Section 1.2. Additionally, Lines 179-185 should be relocated to Section 1.9, which appears to be dedicated to viral vectors, instead of mentioning these vectors early without adequate introduction.

Response: In line 143, we have removed the term non-viral vector (now line 173). Additionally, the text in lines 179–185 has been revised to strengthen the connection between viral vectors and LNPs. We now contextualize the role of LNPs by briefly comparing them with viral vector–based systems, which, despite their long-standing use as effective vaccine delivery tools, present unique challenges (now lines 206, 207…). Furthermore, Section 1.9 on viral vectors has been removed to enhance clarity and maintain a focused discussion on nanoparticle-based carriers.

Comment 14. In Lines 474-487, the manuscript provides only limited mechanistic information for a few NP-based carriers (References 78 and 79). Some NP-based carriers also require additional surface modification to promote intracellular delivery, but this is neither mentioned nor connected to the previous content. This section should establish clearer connections between these mechanisms and the various types of NP-based carriers previously described in Sections 1.2-1.9.

Response: We have added Table 1 to provide a clearer and more comprehensive overview of the mechanisms of action for various nanocarriers, along with their surface modifications to promote intracellular delivery as per available in literature. This table integrates and connects the content from Sections 1.2–1.8, addressing the reviewer’s concern and improving the overall clarity and continuity in the manuscript as well.

Comment 15. In Line 590-591, the three barriers (skin barrier, mucosal barrier, and blood-brain barrier (BBB)) should be discussed earlier in Section 2 (Line 466-487). These barriers relate to mechanisms of nanomaterials delivery at the tissue level, whereas the current Section 2 focuses on cellular-level mechanisms.

Response: We have revised the manuscript to discuss the three barriers (skin, mucosal, and blood-brain barrier) earlier in Section 2 (Lines 466–487), providing context for tissue-level mechanisms of nanomaterial delivery. The subsequent content continues to focus on cellular-level mechanisms, ensuring a logical progression from tissue to cellular considerations (now lines 541, 542, ….).

Comment 16. In Line 613-621, update the references. Reference 100 was published in 2005, and more recent research indicates that the nanocarrier designs described may not effectively cross the BBB. For a proper review, rather than presenting single reference information, it is essential to summarize recent progress, especially in this rapidly advancing research field

Response: We have updated the references in Lines 613–621 to include recent studies from 2023–2024, reflecting current understanding of nanocarrier designs and their ability to cross the blood-brain barrier (BBB). The discussion now summarizes recent progress in this rapidly advancing field (now lines 522, 523….).

Comment 17. In Line 623, it references treatment of glioma, which may discuss how vaccine delivery systems could overcome the BBB crossing challenge. In Line 1040 (Section 4.6) discusses brain tumor treatment. However, vaccines primarily serve as preventive measures that activate the immune system to recognize and combat disease, functioning as prophylactic rather than therapeutic interventions. More broadly, cancer vaccines are a form of immunotherapy designed to harness the patient's immune system to target and destroy cancer cells. These vaccines work by training immune cells to recognize specific markers unique to glioblastoma tumors. Loading medicine in nanocarriers to destroy cancer cells is a treatment strategy, not a vaccine approach. The manuscript should clarify the current status of vaccine nano-delivery systems for brain tumors. If these systems are still in early development, the text should identify specific challenges and potential solutions that connect to the treatment approaches mentioned in these sections. This inconsistency with the title persists throughout sections. The tile should be revised or changed.

Response: We sincerely thank the reviewer for this thoughtful comment. We fully agree that vaccines are primarily prophylactic, designed to activate the immune system to recognize and prevent disease and being prophylactic as well as therapeutic. Currently, there are no clinically approved nano-based vaccine strategies specifically targeting brain tumors such as glioblastoma. Our discussion of glioma treatment and brain tumor therapy (Lines 623 and 1040) was intended to highlight emerging nanocarrier-based approaches that may overcome the blood–brain barrier (BBB) and, in the future, enable the development of therapeutic cancer vaccines for brain tumors. Notably, we have included a recent study by Mendez-Gomez et al. (2024), which uses lipid nanoparticles for mRNA delivery against glioblastoma, and the manuscript has been updated accordingly, removing extraneous information and includes minor modification in title.

Comment 18. In Line 1200 (Section 5), the manuscript relies on outdated references (181-185) published between 2010-2012. The text needs a clearer transition to the carrier systems and should explain how these technologies specifically improve particular delivery systems, not just general description of next-generation sequencing, it has shown the recent progress in 2024 in Reference C1.

Response: We have updated Section 5 (Line 1200) to replace outdated references (181–185) with more recent studies, including 2024 developments highlighted in Reference C1. Additionally, we have improved the transition to the discussion of carrier systems and clarified how these technologies specifically enhance delivery platforms (now line 1234……)

Comment 19. In Line 1225 (Figure 6), the figure needs reorganization. The panel under Transcriptomics is illegible due to its small size.

Response: We have reorganized Figure 6, increasing the size of the panel under Transcriptomics to improve legibility and clarity, and ensured that all panels are consistently formatted for better readability (now line 1248…).

Reviewer suggestions: I recommend narrowing or revising the title to focus on either cancer vaccines or neurological disorders specifically. The manuscript should also clarify the distinction between vaccine approaches and treatment strategies.

Response: We sincerely appreciate the reviewer’s valuable suggestion. However, we want to retain coverage of both cancer and neurological disorders to emphasize the unique and emerging applications of nano-based vaccine and delivery systems. Unlike cancer vaccines, the use of nano-delivery platforms for neurological disorders remains underexplored in the literature. Meanwhile, to address the reviewer’s concern, we have clarified vaccine approaches and improved transitions to ensure a coherent discussion across both fields throughout the manuscript. Including neurological disorders highlights the novelty and broad relevance of our review. To better reflect the focus and limit the scope of the manuscript, the title has been slightly modified to “Nanobased Vaccine Delivery Systems: Innovative Therapeutics against Cancer and Neurological Disorders.

Reference
C1
. Guimaraes, L.C., Costa, P.A.C., Scalzo Júnior, S.R.A. et al. Nanoparticle-based DNA vaccine protects against SARS-CoV-2 variants in female preclinical models. Nat Commun 15, 590 (2024).

< !-- notionvc: cfe5acf5-729f-4495-a0e3-1dc1bd074f88 -->

Reviewer 3 Report

Comments and Suggestions for Authors

The presented article “Nano-Based Vaccines and Delivery Systems: Innovative Thera- 2 peutics Against Cancer and Neurological Disorders” is the large comprehensive review of innovative therapeutic approach by  nanoformulated  systems for stimulated immune response. The authors focus their efforts on description of preventive medicine which is known in classical understanding as vaccine  approach. The article opens many items  in  oncology, neurological ailments, aging  that  evoke difficulties in perceiving all literature material at reading. The scope of the subject under consideration is too large to form a clear and full understanding of the entire problem of nanovaccination. The nanovaccination and issue of delivery of nanoformulated immuneactive drugs deserve to be represented by separate particular research articles. Thus the article has such deficiencies that must be eliminated in the final version.

Comment 1.  What of information sources (bibliographic databases, preprint reports, conference proceedings, study registries, register of controlled trials) were considered? What rational design of structural literature search is laid down in the base of review? Do authors applied the modern systematic analysis and data management for conclusion of perspectives of nanovaccine therapeutic  usage.

Comment 2. At present  its advicable to give the primary  definitions of introduced important concepts and explored subjects (nanovaccine, nanoparticle, nano and microemulsion etc). What characteristics except the particle size  principally differentiate the nanovaccine from usual conventional vaccine?

Comment 3. To improve perception, it is necessary to place comparative  tables with specific recipes, essential  characteristics regarding  each nanosubject refered in article sections.

Сomment 4. Comparative analysis of advantages of different classes of nanovaccines  (inorganic, liposomal compositions etc) must be carried out in sections from 1.2 to 1.9.

Comment 5. Its advicable to create separate tables with preclinical results and clinical verification efficiency dates.

Comment 6. The comparative  table data must contain quantitative characteristics of nanovaccine (the size of particles, time of circulation, pharmacokinetics of release loaded boactive agent) delivery and cellular processes.

Comment 7. The self-microemulsifying drug delivery  and protein-based pickering emulsion systems are not reviewed at all.

Comment 7. What FDA-approved nanovaccines are used in current medicine?

Comment 8. Patent status (age) of nanovaccine medicament must be estimated specifically.

Comment 9. The advantages of aerosol inhalation compared with traditional injection in the case of application of nanoformulated vaccine medicaments are needed to discuss in detail.

In summary: the current review may be submitted for publication in Pharmaceuticals after  revision.

Comments on the Quality of English Language

no comment

Author Response

Reviewer # 3

Comments and Suggestions for Authors

The presented article “Nano-Based Vaccines and Delivery Systems: Innovative Thera- 2 peutics Against Cancer and Neurological Disorders” is the large comprehensive review of innovative therapeutic approach by  nanoformulated  systems for stimulated immune response. The authors focus their efforts on description of preventive medicine which is known in classical understanding as vaccine approach. The article opens many items  in  oncology, neurological ailments, aging  that  evoke difficulties in perceiving all literature material at reading. The scope of the subject under consideration is too large to form a clear and full understanding of the entire problem of nanovaccination. The nanovaccination and issue of delivery of nanoformulated immuneactive drugs deserve to be represented by separate particular research articles. Thus the article has such deficiencies that must be eliminated in the final version.

Response: We thank the reviewer for the comment. To better reflect the focus and limit the scope of the manuscript, the title has been slightly modified to “Nano-based Vaccine Delivery Systems: Innovative Therapeutics against Cancer and Neurological Disorders.

Comment 1.  What of information sources (bibliographic databases, preprint reports, conference proceedings, study registries, register of controlled trials) were considered? What rational design of structural literature search is laid down in the base of review? Do authors applied the modern systematic analysis and data management for conclusion of perspectives of nanovaccine therapeutic usage.

Response: As a narrative review, our aim was to provide a comprehensive overview rather than a systematic synthesis or systematic review. We have included a description of our literature search methods in the manuscript for applying structured screening and data management to synthesize the findings.

Comment 2. At present  its advisable to give the primary definitions of introduced important concepts and explored subjects (nanovaccine, nanoparticle, nano and microemulsion etc). What characteristics except the particle size principally differentiate the nanovaccine from usual conventional vaccine?

Response: We have added clear definitions of key concepts, nanotechnology, nanoparticle, and nano-/emulsions, at the first instance of their introduction in the manuscript. Additionally, we have discussed the principal characteristics that differentiate nano-based vaccines or nano-based carriers.

Comment 3. To improve perception, it is necessary to place comparative tables with specific recipes, essential characteristics regarding  each nanosubject refered in article sections.

Response: We have added comparative Tables 1 and 2 summarizing each nanosubject discussed in the manuscript, including specific formulations, essential characteristics, and relevant references. These tables enhance clarity and allow readers to easily compare and understand the properties and applications of different nano-based vaccine and nanoparticle platforms.

Сomment 4. Comparative analysis of advantages of different classes of nanovaccines  (inorganic, liposomal compositions etc) must be carried out in sections from 1.2 to 1.9.

Response: We have included and explained a comparative analysis of the advantages of different classes of nanocarriers—such as inorganic, liposomal, polymeric and other formulation in Tables 1 and 2. This analysis highlights their respective strengths, applications, and potential for targeted delivery, providing a clearer understanding of each platform.

Comment 5. Its advisable to create separate tables with preclinical results and clinical verification efficiency dates.

Response: We have added separate Tables 1 and 2 summarizing preclinical and clinical results, including study outcomes, to provide a clear and organized overview of nano-based vaccine performance.

Comment 6. The comparative  Table data must contain quantitative characteristics of nanovaccine (the size of particles, time of circulation, pharmacokinetics of release loaded boactive agent) delivery and cellular processes.

Response: We have included the comparative Tables 1 and 2 to include quantitative characteristics of nano-based vaccines, such as particle size, circulation time, release pharmacokinetics of the loaded bioactive agents, delivery efficiency, and associated cellular processes, providing a more detailed and informative comparison across different platforms based on available information.

Comment 7. The self-microemulsifying drug delivery and protein-based pickering emulsion systems are not reviewed at all.

Response: We have now included a discussion of self-microemulsifying drug delivery systems (SMEDDS) and protein-based Pickering emulsion systems in the appropriate section of the manuscript. Although the current literature lacks studies specifically evaluating these systems as carriers for cancer and neurological disorder vaccines, we have discussed available information that how they could be utilized in future research.

Comment 8. What FDA-approved nanovaccines are used in current medicine?

Response: Several vaccines employing nano-based delivery systems for cancer and neurological disorders are currently in clinical trials. Relevant details have been summarized in Tables 1.

Comment 9. Patent status (age) of nanovaccine medicament must be estimated specifically.

Response: We have included a summary of the patent status of key nanovaccine formulations in the Table 1.

Comment 10. The advantages of aerosol inhalation compared with traditional injection in the case of application of nanoformulated vaccine medicaments are needed to discuss in detail.

Response: We thank the reviewer for highlighting the potential importance of aerosol/inhalation delivery routes. While aerosol inhalation offers notable advantages such as non-invasive administration, rapid mucosal immune activation, potential for dose-sparing, and improved patient compliance, while our current review primarily focuses on nano-based delivery systems in cancer and neurological disorder vaccines. The majority of the published studies in this field have employed intravenous (IV), subcutaneous (SC), intradermal, or intratumoral administration routes, which are directly relevant to the vaccines under discussion and have been summarized in Tables 1 and 2.

In summary: the current review may be submitted for publication in Pharmaceuticals after revision.

Round 2

Reviewer 2 Report

Comments and Suggestions for Authors

The second manuscript shows significant improvement; however, additional revisions are necessary before publication. Please address the following suggestions:

  1. In Line 39-40, cancer vaccines are still considered a relatively new medical treatment compared to established major cancer treatment methods (chemotherapy, radiotherapy, and surgical excision) as noted in Ref 4. Since neither Ref 3 nor Ref 4 refer to cancer vaccines as "conventional methods," it would be better to either define "conventional vaccines" earlier in the text or revise the terminology to avoid confusion.
  2. In Line 84-86, Ref 15 cannot support the claim regarding the size range of 50-150 nm. Please assign the appropriate reference, as the NP size was already defined in Line 73-75. Alternatively, consider removing the redundant size range from the sentense.
  3. In Line 104-105, what is histocompatibility complex – I? Add a brief description in an earlier paragraph.
  4. In Line 134-138, avoid emphasizing "without time limitations." Based on the current search methodology, additional references should be included, along with information about clinical status.
  5. In Line 249, Fig 1B does not accurately represent all information from Ref 43. The reference describes two different spatial targeting mechanisms of NPs: (1) passive targeting and (2) active targeting. Additionally, it explains that actively targeted NPs can be utilized in applications where drug release occurs either extracellularly or intracellularly.
  6. In Line 401-402, it's inappropriate to make sweeping claims that "inorganic nanoparticles are hydrophilic, biocompatible, nontoxic and incredibly stable" without specifying the types of inorganic NPs. For instance, Ag nanoparticles easily oxidize without proper surface protection. In Line 402, the reference to magnetic properties is irrelevant for most nanoparticle carrier systems discussed. Lines 403-404 repeat the same generic features for other NP systems, raising the question of why these characteristics are specifically highlighted here. While Reference 71 might describe various inorganic NPs comprehensively, this manuscript doesn't introduce enough inorganic NP materials to support such broad characterizations.
  7. In Line 512, the statement should be revised to be more conservative, as even in Ref 90, the authors did not claim applicability to all types of NPs, because it simply isn't applicable to all cases.
  8. In Line 853-862 (Figure 4), revise the sub-labels in panel B to use lowercase letters (a, b, c, d, e, and f) in the figure legend. Provide more detailed descriptions for each sub-label. Additionally, some labels appear to be duplicates and need correction.
  9. In Line 1251 (Figure 6), what does the right bottom panel with t-SNE2 y-axis and various colored dots represent? The figure legend should provide more information, especially since the body text lacks sufficient explanation of this visualization, not including too much information.
  10. In Line 1278-1279 (Tables 1 and 2), please specify the cancer type in the last column under "Disorder type." Alternatively, if these are intended as universal solutions applicable to multiple cancer types, please indicate this clearly.

< !-- notionvc: 94727be9-9d7a-4aa6-9425-8d4d6e06566a -->

Author Response

Dear Editor,

We extend our sincere gratitude to the 2nd Reviewer for his valuable insights and feedback for our manuscript. We have carefully addressed each of the comments provided by the Reviewer and have revised the manuscript accordingly. Our responses to the comments are detailed below and highlighted in red throughout the revised manuscript for ease of reference

Reviewer # 2

Reviewers' comments:

Comments and Suggestions for Authors

The second manuscript shows significant improvement; however, additional revisions are necessary before publication. Please address the following suggestions:

Comment 1. In Line 39-40, cancer vaccines are still considered a relatively new medical treatment compared to established major cancer treatment methods (chemotherapy, radiotherapy, and surgical excision) as noted in Ref 4. Since neither Ref 3 nor Ref 4 refer to cancer vaccines as "conventional methods," it would be better to either define "conventional vaccines" earlier in the text or revise the terminology to avoid confusion.

Response: Thank you for the insightful comment. We agree that “conventional vaccines” could be confusing in this context, as cancer vaccines remain a developing therapeutic approach compared to standard cancer treatments such as chemotherapy, radiotherapy, and surgery. We have revised the sentence for better terminology accordingly to read: “However, current cancer vaccine approaches often fail to elicit sufficiently strong and targeted immune responses [3, 4] …….” (lines 39-40).

Comment 2. In Line 84-86, Ref 15 cannot support the claim regarding the size range of 50-150 nm. Please assign the appropriate reference, as the NP size was already defined in Line 73-75. Alternatively, consider removing the redundant size range from the sentense.

Response: We agree that the size range (50–150 nm) was already defined earlier in Lines 73–75, and the cited reference [15] does not directly support it here. Therefore, we have removed the redundant size range to ensure accuracy as per suggested.

Comment 3. In Line 104-105, what is histocompatibility complex – I? Add a brief description in an earlier paragraph.

Response: We have added a brief explanation of the major histocompatibility complex class I (MHC-I) in an earlier paragraph to improve clarity. The revised text now reads lines 72-74.

Comment 4. In Line 134-138, avoid emphasizing "without time limitations." Based on the current search methodology, additional references should be included, along with information about clinical status.

Response: We have revised the text to avoid emphasizing “without time limitations” to clarify the inclusion criteria (now lines 140-142).

Comment 5. In Line 249, Fig 1B does not accurately represent all information from Ref 43. The reference describes two different spatial targeting mechanisms of NPs: (1) passive targeting and (2) active targeting. Additionally, it explains that actively targeted NPs can be utilized in applications where drug release occurs either extracellularly or intracellularly.

Response: Fig. 1B was intended to provide a simplified conceptual overview of polymeric nanoparticles encapsulating biomolecules (e.g., antigens) and their subsequent delivery to target cells, including intracellular release. To avoid any misrepresentation of Ref. 43, we have removed this reference and ensured that the figure is now supported by appropriate sources or references that specifically illustrate nanoparticle encapsulation and cellular uptake. The figure is meant to convey the general principle rather than the full spectrum of spatial targeting mechanisms (lines 251-252).

Comment 6. In Line 401-402, it's inappropriate to make sweeping claims that "inorganic nanoparticles are hydrophilic, biocompatible, nontoxic and incredibly stable" without specifying the types of inorganic NPs. For instance, Ag nanoparticles easily oxidize without proper surface protection. In Line 402, the reference to magnetic properties is irrelevant for most nanoparticle carrier systems discussed. Lines 403-404 repeat the same generic features for other NP systems, raising the question of why these characteristics are specifically highlighted here. While Reference 71 might describe various inorganic NPs comprehensively, this manuscript doesn't introduce enough inorganic NP materials to support such broad characterizations.

Response: We have revised the statements to specify types of inorganic nanoparticles (e.g., silica, gold, iron oxide) and clarified that properties such as hydrophilicity, biocompatibility and stability depend on proper surface modification. Irrelevant mentions of magnetic properties have been contextualized for nanoparticles where applicable (lines 404-407).

Comment 7. In Line 512, the statement should be revised to be more conservative, as even in Ref 90, the authors did not claim applicability to all types of NPs, because it simply isn't applicable to all cases.

Response: We have revised the statement to be more conservative, clarifying that not all nanoparticles exhibit the same transdermal delivery capabilities. The revised text emphasizes that the findings from Gupta and Riaz (2017) apply specifically to dodecane thiol–coated hydrophobic gold nanoparticles, and similar mechanisms may be relevant for certain other nanocarriers, rather than generalizing to all nanoparticles (lines 506…….).

Comment 8. In Line 853-862 (Figure 4), revise the sub-labels in panel B to use lowercase letters (a, b, c, d, e, and f) in the figure legend. Provide more detailed descriptions for each sub-label. Additionally, some labels appear to be duplicates and need correction.

Response: The sub-labels in panel B have been revised to use lowercase letters (a, b, c, d, e, and f) for consistency in the figure legend. In addition, more detailed descriptions have been added for each sub-label to enhance clarity and accuracy. The duplicate labels have also been corrected accordingly (lines 864....).

Comment 9. In Line 1251 (Figure 6), what does the right bottom panel with t-SNE2 y-axis and various colored dots represent? The figure legend should provide more information, especially since the body text lacks sufficient explanation of this visualization, not including too much information.

Response: Additional clarification has been added to the figure legend to describe the right bottom panel. The revised legend now includes the following sentence: “The right bottom panel represents a t-distributed stochastic neighbor embedding (t-SNE) plot illustrating single-cell transcriptomic clustering based on gene expression profiles, where each colored dot corresponds to a distinct immune cell subtype.......” This addition provides sufficient context for the visualization without overloading the legend with excessive detail (lines 1263......).

Comment 10. In Line 1278-1279 (Tables 1 and 2), please specify the cancer type in the last column under "Disorder type." Alternatively, if these are intended as universal solutions applicable to multiple cancer types, please indicate this clearly.

Response: The cancer types have now been specified in the “Disorder type” column of Tables 1 and 2 (lines 1276 -1295).

Reviewer 3 Report

Comments and Suggestions for Authors

My respect for your instant reaction.

Author Response

Thank you very much for your review.